# Two-gigapascal-strong ductile soft magnets

Liuliu Han ®[1] ✉, Nicolas J. Peter ®[2], Fernando Maccari ®[3], András Kovács ®[4], Jin Wang[2], Yixuan Zhang[3], Ruiwen Xie[3], Yuxiang Wu[1], Ruth Schwaiger ®[2], Hongbin Zhang[3], Zhiming Li ®[5], Oliver Gutfleisch[3], Rafal E. Dunin-Borkowski ®[4] & Dierk Raabe ®[1]

Soft magnetic materials (SMMs) are indispensable for electromechanical energy conversion in high-efficiency applications, but they are exposed to increasing mechanical loading conditions in electric motors due to higher rotational speeds. Enhancing the yield strength of SMMs is essential to prevent the degradation in magnetic performance and failure from plastic deformation, yet most SMMs have yield strengths far below one gigapascal. Here, we present a multicomponent nanostructuring strategy that doubles the yield strength of SMMs while maintaining ductility. We introduce morphologically anisotropic nanoprecipitates through dislocation-driven precipitation induced by preceding deformation during heat treatment in an iron–nickel–cobalt–tantalum material. With all dimensions of the precipitates below the magnetic domain wall width, we achieve a high precipitate number density with a large specific surface area, small interprecipitate spacing, and high lattice mismatch, which impede dislocation glide and strengthen the material. Both the matrix and precipitates are ferromagnetic, yielding a high magnetic moment. This nanostructuring approach offers a pathway to two-gigapascal-strong ductile SMMs with moderately increased coercivity that can be tolerated in exchange for significantly improved mechanical performance for sustainable electrification.

Soft magnetic materials (SMMs) exhibit rapid magnetic flux variations in response to changes in external magnetic fields. Electrification using SMMs in energy conversion applications is the fastest and most sustainable approach for counteracting climate change and solving the global energy crisis. Minimization of coercivity is crucial for reducing magnetic hysteresis losses in the form of dissipative heat during high-frequency magnetization and demagnetization. At the same time, increasing demand is required for SMMs with high yield strength and good ductility that can withstand harsh and dynamic mechanical loads in electrical motors and high-speed flywheel energy storage systems[1,2]. The yield strength is the stress at which irreversible plastic deformation starts. Thus, loading stresses must remain below this value for safe

and efficient operation of magnetic parts; otherwise, high-speed electrical machines and their rotating parts undergo irreversible dimensional changes[3]. Also, SMMs with good ductility are needed for manufacturing flexibility and damage tolerance, enhancing the lifespan and sustainability of the products[4]. If the yield strength of the material is too low, a build-up of inelastic deformation carriers will impede the free motion of magnetic domain walls, thus deteriorating the soft magnetic properties[2]. Realizing a high yield strength requires the prevention of dislocation nucleation and motion through microscopic backstresses exerted by atomic-scale lattice distortion and lattice defects. This effect, termed mechanical strengthening[5], results in a fundamental material design challenge because the internal stress

[1]Max Planck Institute for Sustainable Materials, Max-Planck-Straße 1, 40237 Düsseldorf, Germany. [2]Institute of Energy and Climate Research (IEK-2), Forschungszentrum Jülich, 52425 Jülich, Germany. [3]Institute of Materials Science, Technical University of Darmstadt, 64287 Darmstadt, Germany. [4]Ernst Ruska-Centre for Microscopy and Spectroscopy with Electrons and Peter Grünberg Institute, Forschungszentrum Jülich, 52425 Jülich, Germany. [5]School of Materials Science and Engineering, Central South University, 410083 Changsha, China. ✉e-mail: l.han@mpie.de

fields of lattice defects can pin magnetic domain wall motion[6], thereby destroying the material's soft magnetic properties. However, conventional microstructural strengthening measures often result in embrittlement and lead to undesired domain wall interactions. Coercivity and hysteresis-related energy losses may increase, and soft magnetic performance is lost[7].

Attempts have been made to address this challenge by introducing and tuning various types, sizes, and coherency states of second-phase precipitates into conventional SMMs[8] and multicomponent-based soft magnets[9–12]. Coherent cuboidal precipitates of well-tuned size can result in low magnetic domain wall pinning. However, the increase in attainable strength has proven to be limited to values far below one gigapascal[10]. This is because the strengthening effect of coherent precipitates by dislocation shearing increases with size but cannot exceed the Orowan stress, at which dislocations circumvent precipitates[5]. In contrast, incoherent precipitates are nonshearable obstacles for dislocations. Yet, their large interfacial energy usually leads to coarse micron-sized precipitates that can strongly pin the motion of magnetic domain walls, thus losing soft magnetic features[13,14].

We propose a strategy to resolve this dilemma through a nanostructuring approach involving morphologically anisotropic nanoprecipitates. The synergistic effects of the morphology, distribution, size, and composition of the precipitates are achieved by the following mechanisms: (a) The anisotropic morphology increases the surface area-to-volume ratio and lattice mismatch, thereby strengthening the material via mechanical interactions with dislocations. (b) A finer size distribution reduces the inter-precipitate spacing, enhancing material strength. (c) The micro-magnetically relevant length scales of the precipitates are kept below the magnetic domain wall width, minimizing interference with domain wall motion and thus reducing magnetic losses[15]. (d) High concentrations of ferromagnetic elements with the highest possible magnetization and moderate costs are used. (e) We address the design challenge of triggering coherent, ferromagnetic, and anisotropic nanoprecipitates under thermodynamically nonequilibrium conditions while suppressing capillary-driven coarsening and avoiding the transition to incoherency.

These considerations have led us to a class of non-equiatomic iron−nickel−cobalt−tantalum multicomponent alloys (MCAs) with ample compositional adjustment options for magnetic, solid solution and precipitation tuning. The first three elements form a ferromagnetic matrix that provides high magnetization, while the fourth element triggers the formation of ordered precipitates. To achieve morphologically anisotropic and coherent nanoprecipitates with a fine distribution, we introduce microbands via plastic deformation along the {111} crystallographic planes to control the distribution density and morphology of the precipitates. The microbands contain a high line density of planar dislocations, which promotes rapid nucleation. Short-term isothermal annealing (1073 K, 10 min) is conducted to trigger the precipitates, whose longitudinal edges lie along the crystallographic slip planes of the face-centered cubic (fcc) matrix. This behaviour, which results from dislocation-assisted nucleation from a quenched supersaturated solid solution matrix, promotes the formation of precipitates with a coherent interface in a thermodynamic nonequilibrium state. This enables rapid nucleation and leads to a high precipitate number density and small interprecipitate spacing. We also observed another type of partially coherent precipitate in a minor fraction with similar chemical compositions and morphologies but different structures coexisting with these coherent precipitates. They are formed successively during the evolution toward a thermodynamically more stable state with a continuously decreasing free energy. An extended isothermal treatment (1073 K, 60 to 300 min) tends to form incoherent precipitates until the equilibrium state is attained. The formation of incoherent phase boundaries and coarsening of precipitates result in an undesired decrease in strength and an increase in coercivity, as described below.

## Results and discussion

### Microstructural analysis

To enhance the mechanical interaction between precipitates and dislocations, we designed an $Fe_{35}Co_{30}Ni_{30}Ta_5$ (at.%) alloy system using the thermodynamic simulation software Thermo-Calc in conjunction with the High Entropy Alloy database (Supplementary Fig. 1). This system aims to form a supersaturated solid solution single-phase structure by solution treatment (1473 K) and a second phase with a different crystal structure by annealing (773−1173 K). The target of these two phases is to obtain a strong ferromagnetic solid solution matrix and an intermetallic phase with different crystal symmetries or anisotropic interfacial energies with the potential to assume an anisotropic morphology. We synthesized the base material referred to as "B-MCA" with a single-phase fcc structure by using conventional casting, hot rolling, and homogenization at 1473 K, as confirmed via X-ray diffraction (XRD) analysis (Supplementary Fig. 2). The B-MCA was cold rolled for 60% thickness reduction to introduce a high line density ($3.2 \times 10^{15} \cdot m^{-2}$) of planar dislocations on crystallographically aligned {111}$_{fcc}$ planes as potential nucleation sites, as shown in Fig. 1a; these locations were recorded using electron channelling contrast (ECC) imaging. The resulting alloy has a single fcc structure with elongated grains along the cold rolling direction and is termed D-MCA, where "D" stands for cold "deformed". No obvious elemental segregation was observed from the microscale down to the near-atomic scale (Supplementary Fig. 3), indicating that the material is chemically homogeneous in this state.

We initiated nanoprecipitation during a thermally activated short-term process by annealing the D-MCA material at 1073 K for 10 min. The resulting alloy contains a high number density ($\sim(2.32 \pm 0.34) \times 10^{22} m^{-3}$) and volume fraction ($\sim$38%) of morphologically anisotropic nanoprecipitates (53 nm in length, 8 nm in width) is termed A-MCA, where "A" stands for "anisotropic", as shown in high-resolution ECC images (Fig. 1b). This approach provides a small effective planar inter-precipitate spacing ($26.1 \pm 6.1$ nm) and a large specific surface area ($3.3 \times 10^8 m^{-1}$). The precipitates are distributed homogeneously in the fcc matrix with an average size of $101 \pm 49$ μm and containing an {011} < 211> texture and deformation-induced twinning (Supplementary Figs. 2–4). The longitudinal edges of these precipitates are parallel to <111>$_{fcc}$ (Supplementary Fig. 4c). Scanning transmission electron microscopy (STEM) high-angle annular dark-field (HAADF) imaging reveals some precipitates are in long-range-ordered hexagonally close-packed (hcp) atomic arrangements (orange frame), and are partially coherent with the disordered matrix (green frame) (Fig. 1c). Interestingly, precipitates in an L1$_2$ (ordered fcc) structure and are fully coherent with the matrix are also identified (Supplementary Fig. 5). The L1$_2$ nanoprecipitates are considered to constitute the major fraction of the precipitates. This is because no diffraction peaks or superlattice diffraction spots were observed for the hcp structure from the XRD analysis and selected area electron diffraction (SAED) measurements (Supplementary Figs. 2, 6). The anisotropic morphology of the precipitates can be attributed to the anisotropic interfacial energies of the precipitates. More specifically, the interfacial energies of the precipitates are 0.11 and 0.64 J m$^{-2}$ for the (111) and (100) planes, respectively, according to density functional theory (DFT) calculations (see Methods). The formation of precipitates with similar morphologies but different structures (fcc, hcp) during kinetic transient might be due to the varying nucleation conditions at different dislocation configurations[16]. Atom probe tomography (APT) analysis (Fig. 1d) reveals the chemistry of the nanoprecipitates and the matrix using a chemical composition threshold of 7 at.% Ta for the precipitate-matrix interface. Fe is depleted from 39.2 at.% in the matrix to 11.0 at.% in the precipitates, while Ta partitions strongly from the

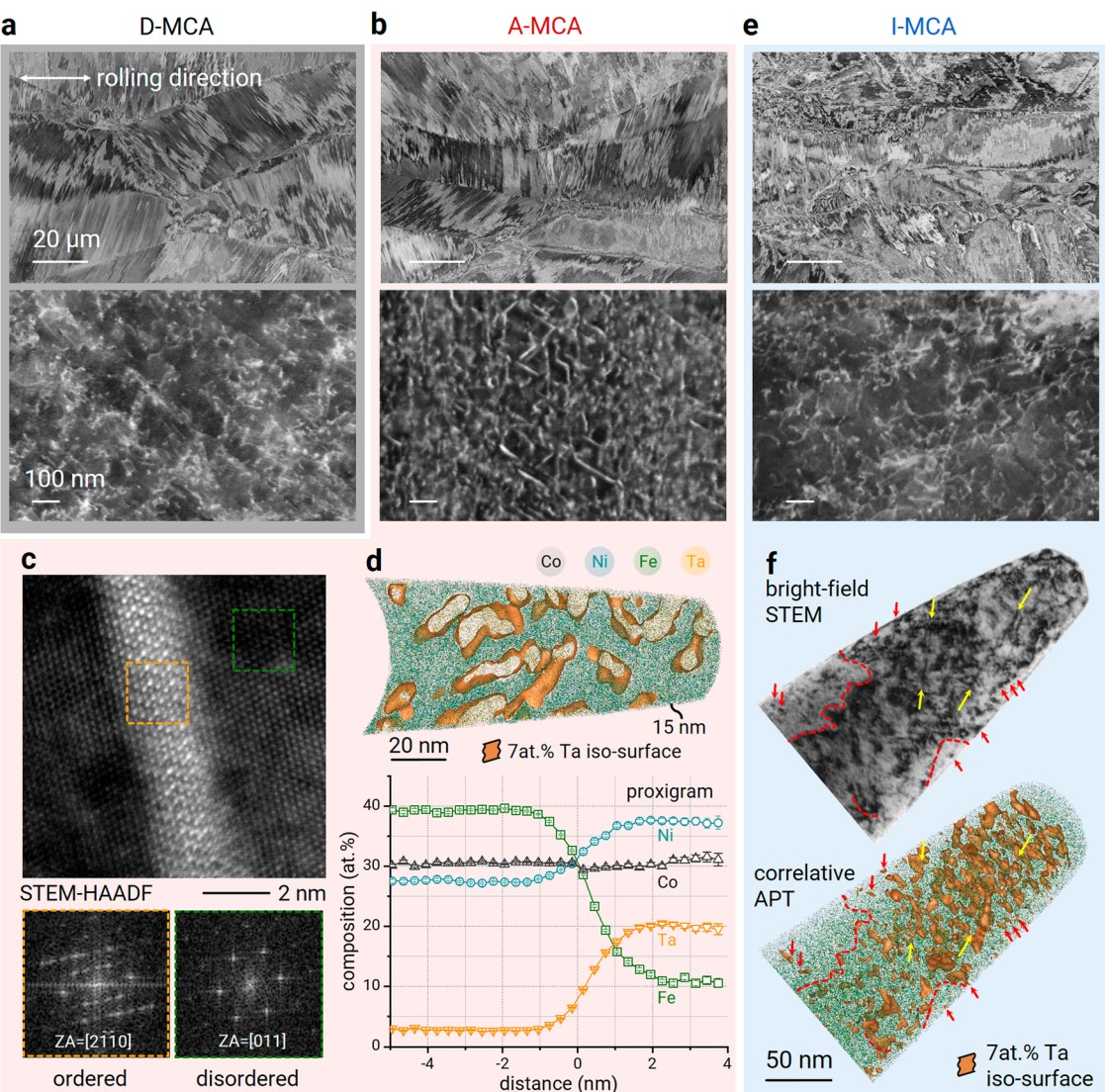

**Fig. 1 | Tuning of the precipitate morphology in the Fe$_{35}$Co$_{30}$Ni$_{30}$Ta$_5$ (at.%) MCA.** ECC images showing an elongated grain structure along the cold rolling direction in the **a** D-MCA, **b** A-MCA, and **e** I-MCA. The lower-magnification images show high microstructural similarity (upper images). The higher magnification images show different features in the grain interior at the submicron scale (lower images). The cold rolling direction is horizontal to the image plane (white arrows). **c** Atomic-resolution STEM HAADF micrograph and corresponding FFT patterns generated from the morphologically anisotropic precipitates (ordered, orange frame) and fcc matrix (disordered, green frame) in the A-MCA. **d** 15-nm-thick slice extracted from an APT measurement showing plate-like precipitates. The corresponding proximity diagram provides the compositions of the precipitate and the matrix. The error bars are estimated as described in Methods. **f** Correlative STEM-APT analysis of the I-MCA material. Yellow arrows mark visible dislocation lines. The red arrows and dashed lines correlate the APT measurements with the corresponding TEM observations. All Ta-enriched regions in the APT map are shown using 7 at.% Ta isosurfaces.

matrix (2.7 at.%) to the precipitates (20.2 at.%). The compositions of the matrix and precipitates are identified as Fe$_{39.2}$Co$_{30.5}$Ni$_{27.6}$Ta$_{2.7}$ and Ni$_{37.7}$Co$_{31.1}$Ta$_{20.2}$Fe$_{11.0}$ (at.%), respectively.

To understand the formation mechanism of the morphologically anisotropic precipitates, we used an intermediate isothermal heat treatment condition by annealing the D-MCA material at 973 K for 10 min. The resulting alloy is referred to as "I-MCA", where "I" denotes "intermediate". I-MCA is expected to have the same precipitation mechanism as A-MCA, albeit with a lower kinetic factor, as indicated by the phase stability function curve (Supplementary Fig. 1b). The dislocation arrangement in I-MCA resembles that in D-MCA (Fig. 1e), with a lower dislocation line density ($2.8 \times 10^{15}$ m$^{-2}$). We characterized the structures and compositions of the crystal defects at the same position using correlative TEM-APT analysis (Fig. 1f). Two dislocation lines, marked by yellow arrows, are identified as Ta-enriched regions, as revealed by the 7 at.% Ta iso-concentration surface in the APT

reconstruction. The red arrows and dashed lines correlate the APT reconstruction map with the TEM image of the tip, showing that not all crystallographic defects are decorated by elemental segregation or are detected by two different microscopy methods. Preferential segregation of Ta atoms to dislocations can be observed, which constitutes regions with highly distorted atomic configurations at their cores and initiates dislocation-assisted preferred precipitation. Pronounced superlattice diffraction was observed from the ordered fcc crystal lattice in the SAED patterns (Supplementary Fig. 6) using the APT tip for correlative experiments. These observations confirm that the solid-state phase transformation of I-MCA is fundamentally the same as that of A-MCA.

Notably, recrystallization was not observed during isothermal heat treatment, even when the annealing temperature and dwelling time were increased to 1173 K (0.7 times the melting point, Supplementary Fig. 7) and 300 min (Supplementary Fig. 8), respectively. This

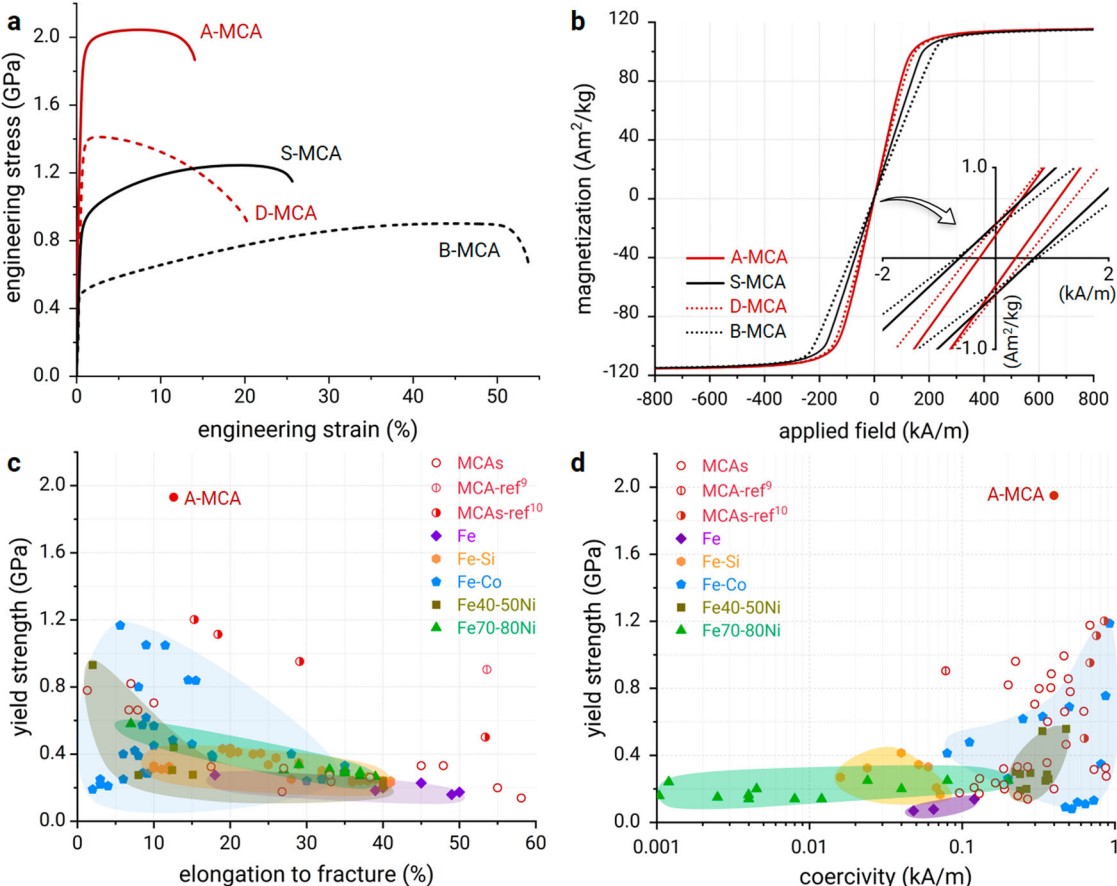

**Fig. 2 | Multi-functional performance of the $Fe_{35}Co_{30}Ni_{30}Ta_5$ (at.%) MCA.** Alloy variants with identical chemical compositions but different microstructural features, including A-MCA, D-MCA, S-MCA and B-MCA materials, are shown. **a** Engineering tensile stress–strain curves. **b** Room-temperature isothermal magnetization measurements. The hysteresis loops were recorded up to $\pm800$ kA·m$^{-1}$ using a field sweeping rate of 1 kA·m$^{-1}$. The inset shows the evolution of the intrinsic coercivity ($H_c$). **c** Ashby map containing a compilation of values of $\sigma_y$–$\varepsilon_f$ for both previously reported and currently developed MCAs for different classes of SMMs, including Fe[58], Fe–Ni[59,60], Fe–Si[19] and Fe–Co[18,61], alongside established MCAs[9,10,13,21,22,62–66]. **d** Ashby plot showing a comparison of the $\sigma_y$–$H_c$ profiles with those of the other soft magnetic materials in **c**.

is because primary static recrystallization is suppressed by (1) the formation of precipitates, which pin grain boundaries, and (2) the high stacking fault energy (SFE) (73.9 mJ·m$^{-2}$; see Methods) of the material, which allows for dislocation double cross slip and mutual annihilation of positive and negative screw components. The isothermal heat treatment must be controlled carefully because prolonging the dwelling time or increasing the temperature promotes the phase transformation to thermodynamically more stable incoherent hcp precipitates (Supplementary Figs. 6c, 8).

To confirm that the planar dislocations introduced by preceding inelastic deformation are essential for triggering anisotropic precipitates, we subjected the B-MCA material to the same isothermal heat treatment (10 min at 1073 K) as A-MCA. The material is termed S-MCA (where "S" denotes "spherical"). STEM and APT analyses (Supplementary Fig. 9) show that the S-MCA material contains coherent spherical precipitates with an average size of ~11.1 ± 4.8 nm and a volume fraction of ~44.3 ± 3.6%.

**Mechanical and magnetic properties**
The introduction of morphologically anisotropic precipitates yields a two-gigapascal-strong ductile soft magnet. Figure 2a shows the tensile engineering stress–strain curve of the A-MCA material. The yield strength ($\sigma_y$) is 1.93 ± 0.02 GPa, and the total elongation is 12.6 ± 0.9%. This high value keeps the material within the elastic region even under severe applied stresses without generating dislocations, thereby preventing irreversible shape changes in

magnetic components during service and undesired interactions between dislocations and magnetic wall motion. Its good ductility allows high manufacturing flexibility and provides damage tolerance. To demonstrate the strength improvement, we present tensile curves for alloy variants with the same bulk chemistry as A-MCA but with different microstructural features: B-MCA (homogenized, no precipitates), D-MCA (cold-rolled, no precipitates), and S-MCA (homogenized and annealed, spherical precipitates). The yield strength of the A-MCA material shows a nearly fourfold increase (~385%) than that of the precipitate-free B-MCA material. The D-MCA material shows no strain-hardening behaviour, meaning it starts necking after reaching the yield point until rupturing. This is because dislocation motion and multiplication (mechanisms that enable plastic ductility) are impeded by the dense dislocation network (~3.2 × 10$^{15}$ m$^{-2}$) introduced by cold rolling. The S-MCA material has a yield strength of less than one gigapascal, similar to $Fe_{32}Co_{28}Ni_{28}Ta_5Al_7$ (at.%), which contains cuboidal precipitates[10]. The A-MCA material maintains a high dislocation density (2.5 × 10$^{15}$ m$^{-2}$) after cold rolling and short-term annealing while simultaneously triggering a high number density (2.3 × 10$^{22}$ m$^{-3}$) of morphologically anisotropic nanoprecipitates, resulting in a high yield strength (1.9 GPa). More importantly, the A-MCA material remains ductile, whereas a significant loss in ductility is generally observed in alloys with high strength. This is regarded as a long-standing fundamental dilemma. The corresponding strain hardening rate (Supplementary Fig. 10) shows a multi-stage work-

hardening capability. Measurements of the impact toughness of the current MCAs, performed using a Charpy impact test, can be found in Supplementary Fig. 11.

Figure 2b shows the magnetic performance of the alloys. The A-MCA material exhibits typical soft ferromagnetic behaviour, with a saturation magnetization ($M_s$) of 115.4 ± 0.2 Am²·kg⁻¹ and a coercivity ($H_c$) of 0.36 ± 0.03 kA·m⁻¹. Compared to our previously designed strong and ductile soft magnet ($Fe_{32}Co_{28}Ni_{28}Ta_5Al_7$ (at.%)[10]) with a relatively low $M_s$ value of 100.2 ± 0.2 Am²·kg⁻¹, the A-MCA material shows a notable increase (~15%) in $M_s$. According to DFT calculations using the exact muffin-tin orbital method based on the unit cell[17], the total magnetic moment of $Fe_{35}Co_{30}Ni_{30}Ta_5$ is 1.50 $\mu_B$/f.u. This value is ~15% higher than that for $Fe_{32}Co_{28}Ni_{28}Ta_5Al_7$ of ~1.32 $\mu_B$/f.u., demonstrating good consistency with the experimental results. Considering that the local magnetic moments of Fe, Co and Ni in $Fe_{35}Co_{30}Ni_{30}Ta_5$ (2.54, 1.57, 0.54 $\mu_B$) are slightly higher than those in $Fe_{32}Co_{28}Ni_{28}Ta_5Al_7$ (2.50, 1.50, 0.47 $\mu_B$), the overall increase in magnetization is attributed primarily to the higher concentration of ferromagnetic elements (Fe+Co+Ni), which is 95 at.% in A-MCA versus 88 at.% in $Fe_{32}Co_{28}Ni_{28}Ta_5Al_7$ (at.%). More importantly, A-MCA has the lowest $H_c$ among the variants of identical chemistry (Supplementary Fig. 12). This situation is achieved by controlling the micromagnetically relevant length scales, specifically the coherency, type, density, and morphology of different microstructural defects, to reduce the pinning of lattice distortion and the effect of the associated internal stress level on magnetic domain wall motion.

## Overview of the multi-dimensional property profiles

To highlight the excellent mechanical-magnetic combination in the A-MCA material containing morphologically anisotropic nanoprecipitates, we compare its room temperature tensile yield strength against elongation at fracture and intrinsic coercivity with those of conventional SMMs ($H_c < 1$ kA·m⁻¹) and established MCAs in the form of Ashby maps (Fig. 2c, d, respectively). The A-MCA material has a high value of $\sigma_y$ with moderate $H_c$. Although a few Fe-Co alloys have relatively high $\sigma_y$, their brittleness and high coercivity preclude their use as mechanically loaded soft magnetic components. One such example is FeCo-2V[18], which has a $\sigma_y$ value of 1.1 GPa but a low ductility value of 5.6% and a high $H_c$ value of 2.9 kA·m⁻¹. It should be noted that although the coercivity of the current A-MCA is higher than that of some conventional SMMs, this is acceptable for applications under high mechanical stresses that prioritize high mechanical strength over minimal coercivity. To highlight the combination of high yield strength and good ductility of the A-MCA material, we compare it with established MCAs in a diagram plotting yield strength against elongation to fracture (Supplementary Fig. 13). This comparison shows that the tensile yield strength of the current A-MCA material outperforms that of all the other MCAs.

For SMMs that target high-frequency applications, the highest possible electrical resistivity ($\rho_e$) is desirable for reducing eddy currents and associated energy losses. MCAs with wide chemical solid solution ranges can be well-tuned for enhanced electrical resistivity due to their high inelastic electron scattering in the distorted lattice. This behaviour is revealed when comparing the $\rho_e$ value of the A-MCA material with those of existing alloy systems (Supplementary Fig. 14). This comparison shows that the $\rho_e$ value of the A-MCA material is higher than the corresponding reference values for all the pure metals, Fe-Si (Si content<6.5 wt.%)[19], Fe-Co[18], comparable to Fe-Ni[20] alloys and established MCAs[21–24].

## Strengthening and toughening mechanisms

To understand the mechanisms that are responsible for the combination of high tensile strength and good ductility in the A-MCA material, we studied the evolution of its deformation substructure using electron backscatter diffraction (EBSD), ECCI and TEM analyses at different straining stages (Fig. 3). The pile-up of geometrically necessary dislocations at grain boundaries can be represented by the kernel average misorientation (KAM) generated during plastic straining. The KAM provides an adequate local measure of inelastic kinematic gradients (lattice curvature) that are accommodated by dislocations of the same Burgers vector sign (Fig. 3a). Microbands aligned along the {111}$_{fcc}$ crystallographic family showed pronounced dynamic refinement (Fig. 3b). This mechanism induces high dislocation passing stresses. It provides continuous strain hardening, which prevents damage initiation. Two types of discontinuous and incoherent grain boundary precipitates enriched in Ta but with different crystal structures and compositions from those inside the grains are observed (Supplementary Fig. 15). The high diffusion coefficient of Ta promotes these two types of precipitates, and enhanced solution depletion occurs along grain boundaries due to interface trapping. For example, the diffusion coefficient of Ta is $4.7 \times 10^{-18}$ m²·s⁻¹, while the corresponding values for Ni, Fe, and Co are $2.6 \times 10^{-18}$, $2.5 \times 10^{-18}$, and $6.3 \times 10^{-19}$ m²·s⁻¹ at 1073 K, respectively. Although strain localization near the grain boundary precipitates results in mechanical weakness and detachment at a later deformation stage (Fig. 3c), the A-MCA material shows a typical ductile fracture, as confirmed by its tensile fracture morphology (Supplementary Fig. 16a, b). This behaviour results from the small volume fraction (0.3%) of the incoherent precipitates, which keeps their influence on damage initiation small.

After the tensile test, we performed TEM analyses to investigate the microstructure near the fracture surface. Multiple parallel mechanical twins with a number density of $(4.2 \pm 0.7) \times 10^{12}$ m⁻² and an average width of 14.7 ± 4.7 nm are observed (Supplementary Fig. 16c). An atomic-resolution STEM micrograph (Fig. 3d) shows twinning in the fcc matrix and the interface between the matrix and precipitates. The FFT patterns are consistent with a $<110>_{twin}$//$<011>_{fcc}$ system. The formation of mechanical twinning is initiated by high tensile stress (~2.0 GPa) during deformation since the stress reaches a critical value for activating twinning (~1.5 GPa; see Methods). It is worth noting that this situation is generally difficult to achieve in bulk fcc materials with SFE values (73.9 mJ·m⁻²) much higher than those of typical SFE regimes required for activating mechanical twinning[25,26] (<~50 mJ·m⁻²), especially under quasistatic tensile loading conditions. This behaviour is attributed to the high strength achieved by interactions between the morphologically anisotropic precipitates and the dense dislocation network, which enables the twinning stress to be reached. This provides an additional plastic deformation mechanism at a later straining stage, further ductilizing and toughening the material.

High strength and good ductility achieved by impeding dislocation motion, and gradual generation of additional dislocations are typically mutually exclusive and conflicting properties. The high strength of the A-MCA material can be attributed to the following strengthening mechanisms. First, ordered morphologically anisotropic nanoprecipitates with a small effective planar inter-precipitate spacing (26.1 nm), high volume fraction (38%), specific surface area ($3.3 \times 10^8$ m⁻¹), and lattice mismatch (1.3%), contribute to a strong precipitation strengthening effect[27]. More specifically, their anisotropic morphology induces a larger specific surface area and associated elastic distortion than precipitates with a spherical shape ($1.9 \times 10^8$ m⁻¹) at the same number density and volume fraction. The specific surface area of our previously designed strong and ductile soft magnet ($Fe_{32}Co_{28}Ni_{28}Ta_5Al_7$ (at.%)[10]), which contains a high volume fraction (55%) and number density ($7.2 \times 10^{20}$ m⁻³) of cuboidal precipitates, is calculated to be $6.6 \times 10^7$ m⁻¹ for comparison. Second, the high number density ($2.5 \times 10^{15}$ m⁻²) of dislocations and associated elastic distortion lead to strong dislocation strengthening. These two effects together contribute to the high additional strength increase of 1.2 GPa (see Methods for details).

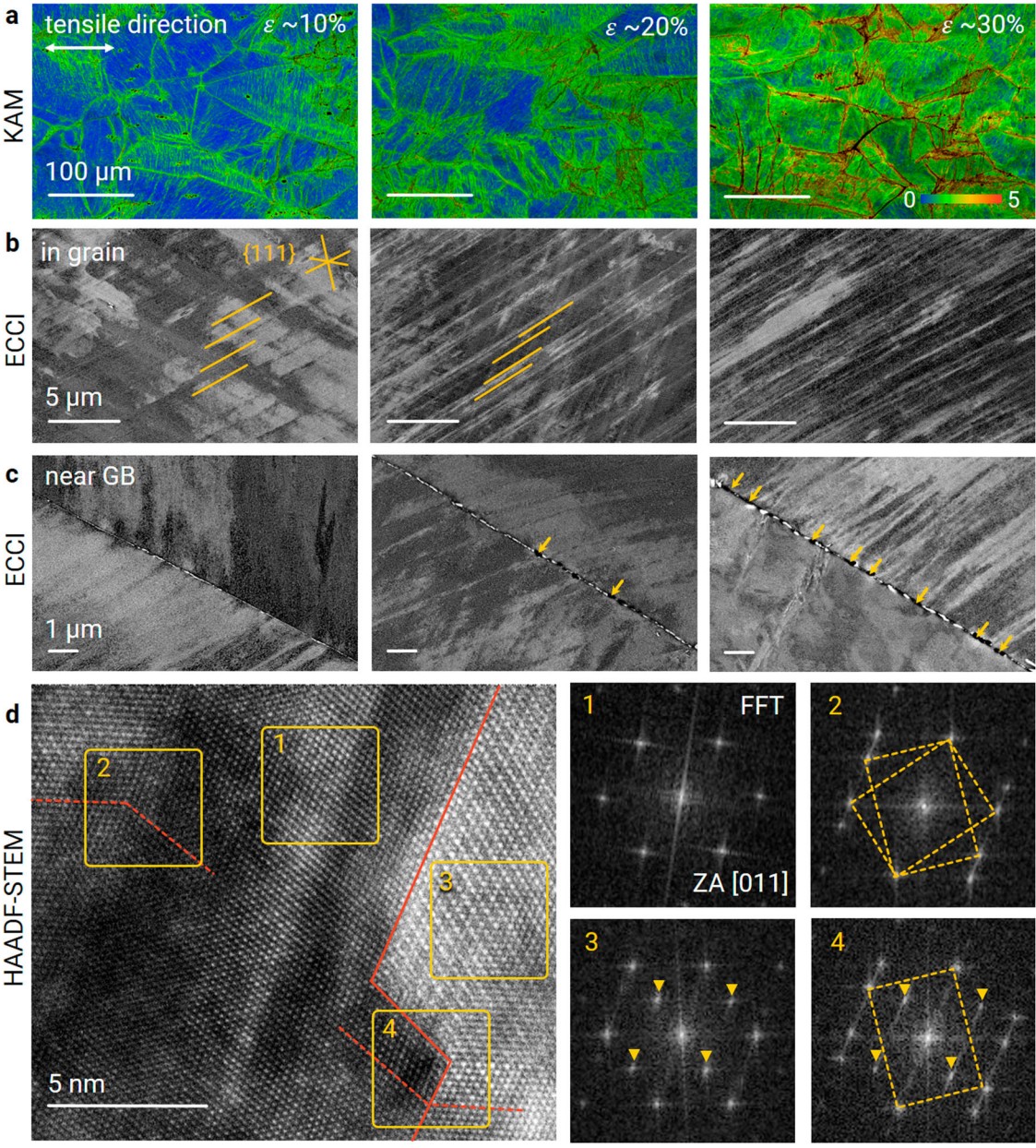

**Fig. 3 | Deformation mechanisms of the A-MCA material. a** Microstructure evolution upon tensile deformation: EBSD-KAM maps showing deformation-induced misorientation features. $\varepsilon$, local deformation. **b** ECC image showing the refinement of microbands during different hardening stages. **c** ECC image showing the detachment of heterogeneous precipitates at grain boundaries. Crack initiation and propagation are marked using orange arrows. **d** Atomic-resolution STEM HAADF analysis showing the interaction of deformation twinning with the matrix and precipitates (left image). FFT analyses (right images) are shown for (1) the disordered fcc matrix, (2) a deformation twin in the fcc matrix, (3) an ordered precipitate, and (4) a deformation twin at the matrix/precipitate interface. ZA, zone axis.

## Correlative magnetic feature characterization and simulation

To understand the mechanisms underlying the magnetic behavior of the A-MCA material, we used multiple experimental and computational techniques to investigate its magnetic features across different length scales (Fig. 4). We conducted correlative magneto-optical Kerr effect (MOKE) microscopy coupled with EBSD analyses (Fig. 4a, b) to characterize interactions between magnetic domains and structural defects during the magnetization process. A variation in the applied magnetic field from −80 to −40 kA·m⁻¹ leads to the preferential nucleation of magnetic domains at grain boundaries and deformation twins[28]. These defects exhibit higher elastic stress fields (~1°) than defect-free grain interiors (~0.2°), as indicated by the geometrically necessary dislocation-mediated crystallographic misorientation in the EBSD-KAM map (Fig. 4a). Magnetic domains with a coarser average size of $25.3 \pm 8.6\ \mu m$ grow unaffected within the grain interiors. In contrast, a significantly refined magnetic domain arranged in stripe-like patterns with an average size of $0.7 \pm 0.1\ \mu m$ is observed between deformation twins (marked as an orange-framed region in Fig. 4a) when the applied field is increased to 40 kA m⁻¹, Fig. 4b. Correlative magnetic force microscopy (MFM) analysis shows the magnetic domain patterns near the twin boundary down to the nanoscale (Fig. 4c). A corresponding topographic profile obtained by atomic force microscopy (AFM) shows that the surface is sufficiently smooth with a height difference less than 10 nm, indicating that the distinctive magnetic domain patterns observed between the twins and the surrounding matrix are attributed primarily to the pronounced

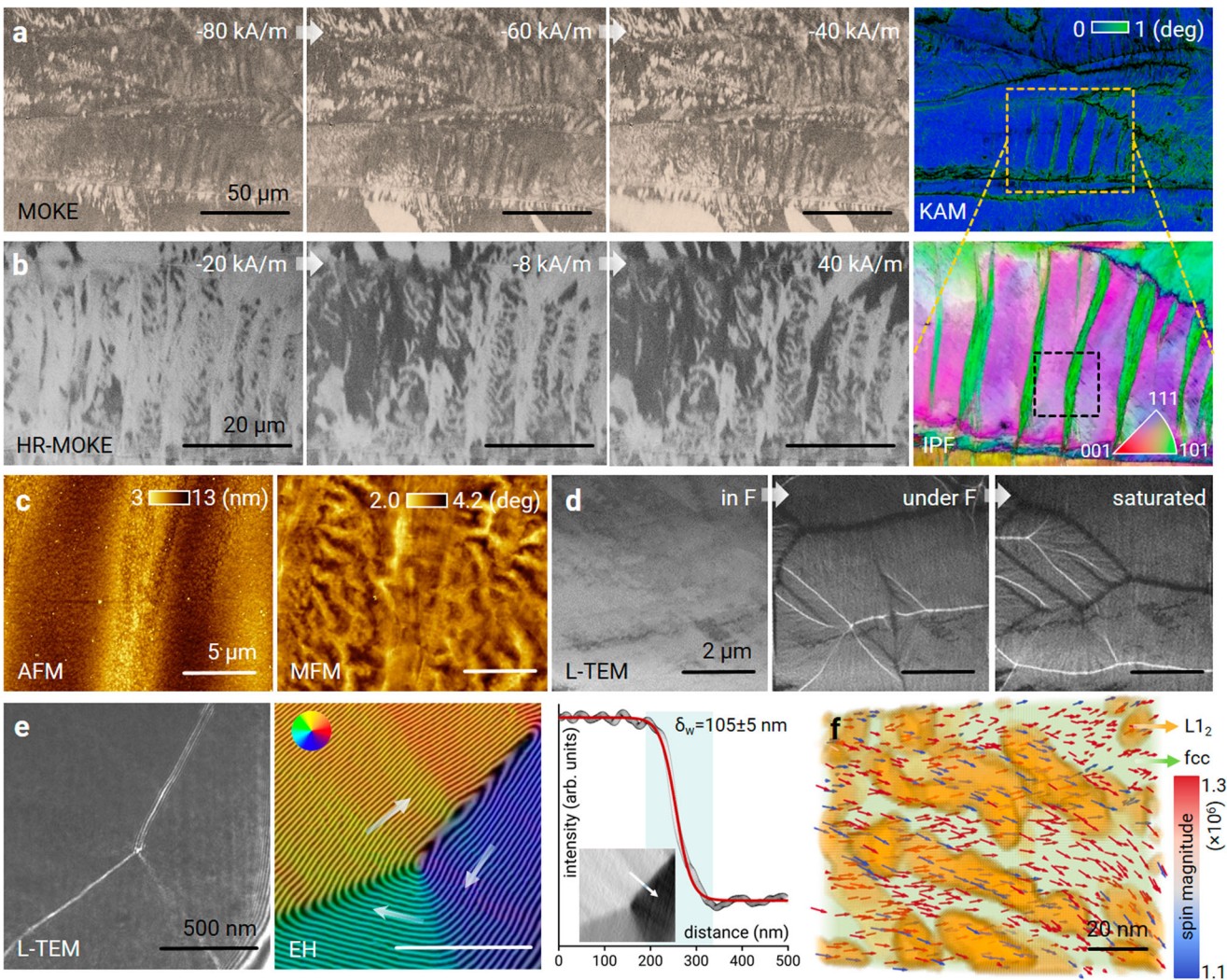

**Fig. 4 | Correlative experimental and computational magnetic feature analysis from the microscale to the nanoscale in the A-MCA material. a** Evolution of the magnetic domain pattern recorded using MOKE microscopy under different applied magnetic fields and a correlative EBSD-KAM map. **b** High-resolution MOKE images and a corresponding EBSD-IPF map recorded from the region marked by the orange dashed frame in **a**. **c** Correlative AFM-MFM images showing magnetic domain features adjacent to the cold rolling-induced twinning region, corresponding to the black dashed frames in **b**. **d** Lorentz TEM analysis. Left and middle: in-focus and underfocus Fresnel images of magnetic domain walls. Right: underfocus Fresnel image recorded after saturating the sample. The defocus used to record each image was 1 mm. **e** Fresnel image (left) and corresponding magnetic induction map measured using off-axis electron holography (middle) of a magnetic flux closure state. The contour lines have a spacing of 2π radians and show the projected in-plane magnetic induction. The magnetic domain wall width $\delta_w$ was determined by fitting a function to the differential of the electron optical phase shift measured using off-axis electron holography (right, inset). **f** Simulated 3D magnetic configurations, highlighting the exchange interaction between the fcc matrix and the morphologically anisotropic nanoprecipitates. The magnetization vector direction and color denote the local magnetization direction and magnitude, respectively.

crystallographic misorientation induced by the preceding cold rolling[29] (Supplementary Fig. 4a). Figure 4d shows the magnetic domain structure in a TEM lamella, as recorded by Fresnel imaging at the magnetic remanence state (0 mT). The alternate black and white lines represent magnetic domain walls. The magnetic domains were observed to reform randomly after saturation using a 1.5 T out-of-plane magnetic field, indicating that the magnetic matrix is intrinsically soft. The magnetic domain pattern exhibited flux closure states, comprising 180° and 90° magnetic domain walls (Fig. 4e). The magnetic domain wall width ($\delta_w$) was quantified to be 105 ± 5 nm at a 180° domain wall by analyzing the differential of the electron optical phase shift measured using off-axis electron holography[30] (Fig. 4f). As the values of $\delta_w$ for conventional SMMs range from 50 nm to a few micrometres[19], the dimensions of the precipitates (53 nm in length and 8 nm in width) are smaller than those of $\delta_w$, meaning that they do not act as strong pinning sites for magnetic domain wall motion. Considering that the

lattice mismatch and associated interfacial elastic distortion can also exert a local pinning effect on magnetic domain wall motion, we conducted three-dimensional micromagnetic simulations by combining microstructural data from APT and TEM measurements with magnetic parameters obtained from DFT calculations (see Methods and Supplementary Fig. 17a). A representative simulation (Fig. 4f) highlights the exchange interaction between the stronger ferromagnetic fcc matrix ($M_s$ ~ 161.6 Am²·kg⁻¹) and weaker ferromagnetic precipitates ($M_s$ ~ 38.7 Am²·kg⁻¹). Such interphase exchange results in negligible pinning because of the relatively weak magnetism of the precipitates, which is the determining effect that provides high strength without sacrificing coercivity. To determine the effect of the precipitates on the overall coercivity, we conducted micromagnetic simulations of the fcc matrix and precipitates in the same configuration (Supplementary Fig. 17b, c). The introduction of weak ferromagnetic anisotropic precipitates in the fcc matrix results in an 18%

decrease in coercivity compared to that of the A-MCA material. The primary reason is the decreased exchange stiffness of the precipitates (Supplementary Table 1), which results in a low external field available for spin rotation. This analysis reveals that the size, morphology, structure, and composition of the precipitates have to be tuned carefully, as these parameters determine their mechanical and physical response. It is worth noting that the coercivity values of the current materials are slightly higher compared to those of conventional SMMs. Nevertheless, the excellent mechanical properties, i.e., high yield strength paired and good ductility make it suitable for applications under severe mechanical stresses where mechanical robustness outweighs the need for minimal coercivity. This tuning can be achieved by balancing the micromagnetically relevant length scales of morphologically anisotropic precipitates (maintaining high mechanical strength) and releasing structural defects (reducing coercivity).

In summary, we designed a bulk multicomponent soft magnetic material with the following mechanical properties: two-gigapascal strength, good ductility (12%), high saturation magnetization (115 $Am^2 \cdot kg^{-1}$), relatively low coercivity (0.36 $kA \cdot m^{-1}$) and high electrical resistivity (62.5 $\mu\Omega \cdot cm$). We realized this concept by engineering morphologically anisotropic and coherent nanoprecipitates with optimized spatial dimensions and patterns, chemical compositions, and coherency stresses. The magnetic properties of the material and the corresponding magnetization mechanism are revealed by using correlative experimental and computational probes across a wide range of length scales. Multiple mechanical strengthening and toughening mechanisms are combined with soft magnetic features, yielding a multi-functional property profile. The strengthening mechanisms have been realized in a manner to not interfere with the magnetic domain wall motion, thus reconciling otherwise mutually exclusive material design targets. The versatile and practically infinite compositional space of such multicomponent alloys and nonequilibrium nanostructuring design concepts are critical for developing energy-efficient SMMs that can endure severe mechanical loading conditions without sacrificing their low hysteresis loss features. Although introducing a high number density and different types of lattice defects results in a moderately higher coercivity, this material design concept provides an approach for highly efficient electromechanical energy conversion applications where mechanical robustness is prioritized over minimal coercivity. It is particularly suited for highly mechanically stressed magnetic parts exposed to severe centrifugal mechanical loads during service or manufacturing, for which conventional SMMs are mechanically too brittle or too soft. Applications lie in the increasing demand for soft magnetic components in electrical vehicle motors and high-speed flywheel energy storage devices, systems exposed to harsh and dynamic mechanical loads.

## Methods

### Material preparation
An alloy ingot with a predetermined nominal composition of $Fe_{35}Co_{30}Ni_{30}Ta_5$ (at.%) was cast in a vacuum induction furnace using high-purity ( > 99.8 wt.%) metallic ingredients in a high-purity argon (Ar) atmosphere for protection. The chemical composition of the as-cast ingot with dimensions of 40 mm (length) × 40 mm (width) × 25 mm (thickness) was measured via wet chemistry analysis and determined to be $Fe_{34.51}Co_{30.32}Ni_{30.23}Ta_{4.94}$ (at.%). The ingot was hot rolled at ~1473 K to achieve an engineering thickness reduction of 50% (from 25 to 12.5 mm) and then homogenized at the same temperature for 10 min under Ar, followed by water quenching. The surface 1 mm of the hot-rolled sheet was removed to avoid the potentially detrimental effect of the oxidation layer. The alloy sheets were then cold rolled at room temperature to achieve an engineering thickness reduction of 60% (from 10.5 to 4.2 mm). Materials containing different dimensions, shape factors, compositions and types of precipitates were realized with

subsequent isothermal heat treatments. Details of the sample identification, thermomechanical processing and microstructural features are summarized in Supplementary Table 2.

### Structural analysis methods
X-ray diffraction (XRD) analysis was carried out with an X-ray diffractometer (D8 Advance A25-X1) using Co Kα radiation (λ = 1.78897 Å) at 35 kV and 40 mA. Quantitative phase analysis was performed with Rietveld simulation using Bruker software (TOPAS5). Scanning electron microscopy (SEM) was carried out at 30 kV using a focused ion beam scanning electron microscope (Carl Zeiss Sigma, Germany). Electron channeling contrast imaging (ECCI) and electron backscatter diffraction (EBSD) characterizations were conducted using a high-resolution (HR) field emission electron microscope (Carl Zeiss Merlin, Germany). The electron channelling pattern (ECP) for controlled ECC imaging was simulated using tools for orientation determination and crystallographic analysis software (TOCA) and EBSD orientation maps. High-resolution scanning transmission electron microscopy (STEM) was carried out using a probe aberration-corrected transmission electron microscope at 300 kV (FEI Titan Themis, United States). A convergence semiangle of 23.8 mrad and collection semiangles of 73–200 mrad were used to achieve Z-contrast characteristics in high-angle annular dark-field (HAADF) imaging mode. Corresponding energy-dispersive X-ray spectroscopy (EDS) observations were performed using an EDS detector (Thermo Fischer Scientific's Super-X windowless) at the same acceleration voltage. Three-dimensional elemental distributions at the near-atomic scale were investigated using atom probe tomography (APT) performed with a local electrode atom probe (Cameca Instruments, Inc., LEAP 5000 XR) in laser-pulsed mode. Visualization and quantification of APT datasets were conducted using commercial software (AP Suite V6). A pulse energy of 40 pJ, a pulse frequency of 125 kHz and a detection frequency of 1 ion per 100 pulses at a temperature of 60 K were used for APT experiments. Correlative TEM-APT experiments were first performed via a transmission electron microscope (JEOL 2200FS, Japan) at 200 kV using a sharpened atom probe specimen. The error bars in all the APT analyses were calculated by $2\sigma = \sqrt{\frac{C_i(1-C_i)}{N}}$, where $C_i$ is the composition of each solute $i$, $N$ is the total number of atoms of the analysis volume. A cleaning procedure using an etching and coating system (PECS, GATAN, Inc.) was conducted before collecting correlative APT data from the same specimen.

### Magnetic feature analysis methods
Magnetic domains were observed using magneto-optical Kerr effect (MOKE) microscopy (evico magnetics GmbH, Germany). A background image was collected in the saturated state (with an applied magnetic field of 255 $kA \cdot m^{-1}$) as a reference before the measurement. Micrographs recorded in different magnetic fields were enhanced by subtracting the background image using KerrLab software. Magnetic force microscopy (MFM) and atomic force microscopy (AFM) were conducted using an atomic force microscope (Bruker, Dimension icon, United States) equipped with a soft magnetic probe (PPP-MFMR cantilever from NanoSENSORS). Standard dual-pass mode was used. The first mode records the topographic features, while the second mode records information about the magnetic domain structure/stray field using a lift height of 50 nm. The average magnetic domain size ($W_{SD}$) was measured using a stereological method[28]: $W_{SD} = 2L_t/(\pi N_i)$ after examination over a total area of 5 $mm^2$, where $L_t$ is the full test line length and $N_i$ is the number of intersections between the test lines and the magnetic domain wall. All correlative measurements (Kerr, AFM, and MFM) were conducted at room temperature. Magnetic domain structures and corresponding induction maps were measured and visualized using a focused ion beam (FIB)-prepared lamella in an image aberration-corrected transmission electron microscope (FEI, Titan 80-300, United States) operated at 300 kV by switching off the objective

lens to achieve magnetic field-free conditions (Lorentz mode). Fresnel defocus images and off-axis electron holograms were recorded on a direct electron counting detector (Gatan K2 IS, United States) with $4k \times 4k$ pixels. Magnetic induction maps were obtained using a standard fast Fourier transform approach in a custom-made software package[31]. The position of the 180° domain wall was determined from the magnetic induction map (Fig. 4e). The phase shift across the domain wall was measured from the original phase image. Its width was determined from the differential of the recorded phase shift (Fig. 4e, inset). Nonlinear curve fitting (red line) to the differential of the phase shift was performed by using the following equation[30]:

$$y = \pm k \times \tanh\left(\frac{\pi \times (x - x_0)}{w}\right) \qquad (1)$$

where $w$ is the width of the domain wall, $k$ is the amplitude of the recorded signal and $x_0$ is an offset that can be obtained from the fit.

## Mechanical response measurements

Room temperature uniaxial tensile tests were carried out using flat dog-bone-shaped specimens with an initial strain rate of $1 \times 10^{-3} \cdot s^{-1}$. Tensile specimens with dimensions of 20/10 mm (total/gauge length) × 2 mm (gauge width) × 1 mm (thickness) were cut from the alloy sheets along the rolling direction (RD) using electrical discharge machining. The surfaces of the specimens were ground to 1000 grit using SiC paper before the tension tests. The local strain evolution and distribution during tensile testing were recorded using the digital image correlation (DIC) method. Charpy impact tests at room (300 K) and cryogenic (93 K) temperatures were performed using a Zwick/Roell machine (RKP450, Germany) on a Charpy V-notch (CVN) specimen with dimensions of 27 mm (length) × 4 mm (width) × 3 mm (thickness). The specimens were machined along the RD with the 60° notch perpendicular to the rolling plane. The geometry of the specimen is based on a European standard (EN 10045-1). A K-type thermocouple was used to measure the temperature during the test. Nanoindentation experiments were performed using an in situ nanoindenter (FT-NMT04, Femtotools, Buchs, Switzerland) with a diamond Berkovich tip. The frame stiffness and area function of the nanoindenter were calibrated using fused silica as a reference material[32]. The elastic modulus and hardness of the A-MCA material were measured using a continuous stiffness measurement (CSM) method[33]. The strain rate was kept constant at $0.025 \, s^{-1}$ during the test, and the indentation depth increased from 400 to 1500 nm.

## Physical (magnetic and electrical) response measurements

The magnetic and electrical properties of the bulk specimen were evaluated at room temperature using a physical property measurement system (PPMS, Quantum Design, United States). The magnetic response was evaluated using vibrating sample magnetometry (VSM) under open-circuit conditions. The dimensions of the specimens for hysteresis loop measurement were 3 mm (length) × 3 mm (width) × 1 mm (thickness). A step size in field$^{-1/2}$ with an average time of 1 s and one repetition at each field was used to measure the hysteresis loop precisely.

The electrical resistivity was determined using the electrical transport option (ETO) from cuboid specimens with dimensions of 6 mm (length) × 2 mm (width) × 1 mm (thickness). Resistivity values ($\rho_e$) were calculated using the following expression:

$$\rho_e = \frac{RA}{l} \qquad (2)$$

where $R$ is the resistance, $A$ is the average cross-sectional area through which the current is passed, and $l$ is the voltage lead separation. At least three specimens were tested for each magnetic and electrical condition.

## Density functional theory (DFT) calculations

Ab initio calculations were performed using the exact muffin-tin orbital method[17]. The Perdew–Burke–Ernzerhof exchange–correlation functional[34] was applied for self-consistent calculations. Chemical disorders were simulated using the coherent potential approximation[35]. The $s$, $p$, $d$ and $f$ orbitals were included in the basis set for solving the one-electron Kohn–Sham equations, except that the Ta-$4f$ shell was taken as the core state.

The elastic constants and SFE of the MCA materials were evaluated in the ferromagnetic state. Three independent single-crystal elastic constants[36], $C_{11}$, $C_{12}$ and $C_{44}$, were calculated for the matrix and nanoprecipitates. The precipitates were considered to be an ordered (L1$_2$) fcc phase based on the XRD results (Supplementary Fig. 2). A total of 20,000 to 25,000 uniformly distributed $k$-points in irreducible Brillouin zones were sampled to ensure the high accuracy of the calculated elastic constants. A stacking fault in the fcc matrix was configured and calculated by shearing one Burgers vector along the <112> direction in a 9-layer supercell with successive {111} planes. The SFE was obtained by using SFE $= (E^{SF} - E^{fcc})/A$, where $E^{SF}$ and $E^{fcc}$ are the energies of structures with and without a stacking fault, respectively, and $A$ is the planar surface area[37]. $k$-meshes of $17 \times 17 \times 1$ and $13 \times 13 \times 1$ were used for SFE calculations of the fcc and L1$_2$ phases, respectively.

Magnetic anisotropy constants ($K_1$) of the fcc and L1$_2$ phases were calculated based on the magnetic torque[38] using the spin-polarized relativistic Korringa–Kohn–Rostoker (SPR-KKR) method[39]. The torque $T_\varphi(\vartheta = \frac{\pi}{2}, \varphi = \frac{\pi}{8})$ is equivalent to $-K_1/2$[29] for cubic crystal structures. The $K_1$ values for the fcc and L1$_2$ phases were calculated to be 964 and $-2637 \, J \cdot m^{-3}$, respectively. The negative $K_1$ value of the L1$_2$ phase indicates that its magnetization prefers the [111] direction over the [001] direction. The exchange stiffness ($A_{ex}$) of the fcc and L1$_2$ phases was calculated using the exchange parameters by assuming a continuous long wavelength function for the spin distribution in the Heisenberg Hamiltonian[40]. Exchange parameters were obtained using the SPR-KKR method based on KKR Green's function[41]. All the computational parameters used are shown in Supplementary Table 1.

The crystal structures of the nominal bulk chemistry (Fe$_{35}$Co$_{30}$Ni$_{30}$Ta$_5$, at.%), the matrix (Fe$_{39.2}$Co$_{30.5}$Ni$_{27.6}$Ta$_{2.7}$, at.%), and the precipitate (Ni$_{37.7}$Co$_{31.1}$Ta$_{20.2}$Fe$_{11.0}$, at.%, modeled as Ta$_{80.8}$Fe$_{19.2}$(Ni$_{50.3}$Co$_{41.5}$Fe$_{8.2}$)$_3$) in the A-MCA material are constructed based on special quasi-random structures (SQS) technique[42]. The interface structures between the matrix and precipitate with crystalline orientations fcc$_{(100)}$/L1$_{2(100)}$, fcc$_{(110)}$/L1$_{2(110)}$, and fcc$_{(111)}$/L1$_{2(111)}$ are built using the same technique. The compositions for different surface indexes of fcc and L1$_2$ phases are considered identical to the bulk phases (Supplementary Fig. 18). Based on the thickness of the surface structure, the numbers of constructed interface structures of fcc$_{(100)}$/L1$_{2(100)}$, fcc$_{(110)}$/L1$_{2(110)}$, and fcc$_{(111)}$/L1$_{2(111)}$ are six, eight, and six, respectively.

The formation energy of the interface is estimated by:

$$E_f = E_{inter} - n_{Fe}*E_{Fe} - n_{Ni}*E_{Ni} - n_{Co}*E_{Co} - n_{Ta}*E_{Ta} \qquad (3)$$

where $E_{inter}$ is the total energy of the interface and $E_{Fe}, E_{Ni}, E_{Co}, E_{Ta}$ are the elemental energies in their stable phases, respectively, and $n$ is the number of elements constituting the interface structure. All related calculations were performed at the ferromagnetic state using the Vienna Ab initio Simulation Package (VASP) with the projector-augmented wave potentials[43,44]. The generalized gradient approximation (GGA) of Perdew, Burke and Ernzerhof (PBE) was utilized[45]. All structures were relaxed regarding the internal degree of freedom until the force exerted on the atom was less than 0.02 eV/Å with an energy cutoff of 400 eV. The Γ-point calculations were performed with an energy convergence criterion of $10^{-6}$ eV due to the large size of the simulation cell.

The average formation energies corresponding to fcc$_{(100)}$/L1$_{2(100)}$, fcc$_{(110)}$/L1$_{2(110)}$, and fcc$_{(111)}$/L1$_{2(111)}$ are −0.115, −0.097, −0.086 eV/atom, respectively. The fcc$_{(100)}$/L1$_{2(100)}$ orientation is energetically most preferred. In comparison, the matrix bulk fcc and L1$_2$ phases exhibit formation energies of −0.008 and −0.229 eV/atom, respectively, while the nominal fcc phase shows a positive formation energy of 0.010 eV/atom. Therefore, the decomposition of the nominal fcc phase can be understood as a progression toward the formation of phases characterized by enhanced thermodynamic stability, as manifested by the particularly negative formation energy of the matrix L1$_2$ phase.

### Estimation of the critical stress for twinning

The DFT calculations showed that the fcc matrix has a high SFE of ~73.9 mJ·m$^{-2}$. Despite such a high SFE, deformation-induced mechanical twinning was observed in the A-MCA material. We therefore evaluated the critical stress for twinning ($\sigma_{\text{twinning}}$) based on the mean-field model[25,26]:

$$\sigma_{\text{twinning}} = M\left(\frac{\Gamma}{b_p} + \frac{G \cdot b_p}{D}\right) \qquad (4)$$

where $M$ is the Taylor factor for a polycrystalline material. The value of $M$ can be taken as 3.06 based on the Taylor model or 2.7 based on the crystal plasticity finite element model[46]. $\Gamma$ is the SFE (73.9 mJ·m$^{-2}$), $b_p$ is the magnitude of the Burgers vector of the partial dislocation (0.145 nm), and $D$ is the average grain size (101 μm). $G$ is the polycrystal shear modulus calculated using[47]: $G = E/2(1 + \nu)$, where $E$ is the elastic modulus measured from nanoindentation experiments (Supplementary Fig. 19), and the Poisson $\nu$ is taken to be 0.3. $G$ and $\sigma_t$ of the current A-MCA material are calculated to be 88 and 1.38 ~ 1.56 GPa, respectively. Therefore, introducing a high number density of metastable, plate-like precipitates and dislocations plays a significant role in providing enough strength to exceed the required critical stress to trigger twinning during straining, activating multistage straining hardening behaviour, and leading to a 2-GPa yet ductile soft magnetic material.

### Three-dimensional micromagnetic simulations

The three-dimensional magnetic structure was determined from the experimental APT and TEM data by using a self-developed phase segmentation algorithm. Micromagnetic simulations were performed using MuMax3 software[48,49]. The digitalization procedure was inspired by the principles inherent to convolutional networks[50]. The APT reconstruction tip with dimensions of 100 nm × 100 nm × 220 nm was divided into ~1.7 × 10$^7$ simulation blocks, each with a size of 0.5 nm × 0.5 nm × 0.5 nm. The chemical composition of each block was first calculated. The cosine similarities between the compositions of the blocks and phases were subsequently used to determine the phase types of the blocks. This phase determination process may introduce a statistical bias due to the relatively small sizes of individual blocks and possible compositional fluctuations in the APT dataset. Therefore, neighboring blocks were also included to achieve precise phase identification by channelling the compositions of central blocks and adjacent blocks through a kernel matrix. The output was considered the ultimate phase information of the central block. In this way, the phases and their boundaries were determined accurately for micromagnetic simulations. The magnetization and magnetization reversal processes were simulated using finite-difference discretization with spatial and temporal variations. The required parameters, including magneto-crystalline anisotropy, exchange stiffness and saturation magnetization, were obtained via DFT calculations (Supplementary Table 1). The magnetization dynamics were computed at a sweep rate of 0.001 T over ±0.3 T by solving the time-dependent

Landau–Lifshitz–Gilbert (LLG) equation as follows:

$$\frac{\partial \mathbf{m}}{\partial t} = -|\gamma|\mu^0 \mathbf{m} \times \left(H_{ext} + \frac{2K^1}{M_s}(\mathbf{m} \cdot \mathbf{n})\mathbf{n} + \frac{2A}{\mu^{0M_s 2}}\nabla^2 \mathbf{m}\right) + \alpha \mathbf{m} \times \frac{\partial \mathbf{m}}{\partial t} \qquad (5)$$

where $\frac{\partial \mathbf{m}}{\partial t}$ is the rate of change in the magnetization process, $\gamma$ is the gyromagnetic ratio, $\mathbf{m}$ is the magnetization vector, $H_{ext}$ is the external magnetic field, $K_1$ is the magnetic anisotropy constant, $A$ is the exchange stiffness, $\mathbf{n}$ is the direction of easy magnetization, $\mu_0$ is the vacuum permeability, and $\alpha$ is the Gilbert damping constant.

### Estimation of the specific surface area of the precipitates

The specific surface area ($S_{specific}$) of the precipitates was determined from the expression:

$$S_{\text{specific}} = S_{\text{surface}}/V_{\text{surface}} \qquad (6)$$

where $S_{\text{surface}}$ and $V_{\text{surface}}$ are the surface area and volume of the precipitates, respectively. The volume of the individual plate-like precipitates ($V_{\text{plate-like}}$) in the A-MCA material is estimated as follows:

$$V_{\text{plate-like}} = V \cdot f_v/\rho_n \qquad (7)$$

where $V$ is the total volume and $f_v$ and $\rho_n$ are the experimentally measured volume fraction and number density of the precipitates, respectively. $V_{\text{plate-like}}$ is calculated to be $1.64 \times 10^{-23}$ m$^{-3}$. The plate-like precipitates in the A-MCA material are approximated to have rectangular shapes, resulting in the expression:

$$V_{\text{plate-like}} = l \cdot w \cdot t \qquad (8)$$

where $l$ and $w$ are the average length and width of the precipitates, respectively, and $t$ is the third dimension, which is calculated to be 35.9 nm. $S_{specific}$ of the plate-like precipitates is then calculated as follows:

$$S_{\text{specific-plate-like}} = 2\frac{(l \cdot w + w \cdot t + l \cdot t)}{(l \cdot w \cdot t)} \qquad (9)$$

$S_{\text{specific-plate-like}}$ is estimated to be $3.26 \times 10^8$ m$^{-1}$. To determine the high $S_d$ value for the precipitates in the A-MCA material due to their unique plate-like morphology, we calculated $S_{\text{specific}}$ for precipitates with different morphologies by keeping their volume and number density constant. $S_{\text{specific}}$ of precipitates with spherical shapes are estimated to be:

$$S_{\text{specific-spherical}} = \frac{4\pi r^2}{\left(\frac{4}{3}\pi r^3\right)} \qquad (10)$$

where $r$ is calculated to be $1.58 \times 10^{-8}$ m$^{-1}$ and $S_{\text{specific-spherical}}$ is $1.90 \times 10^8$ m$^{-1}$. We also estimated $S_{\text{specific}}$ for precipitates in the previously reported strong and ductile Fe$_{32}$Co$_{28}$Ni$_{28}$Ta$_5$Al$_7$ (at.%) multicomponent soft magnetic material[10] containing a high volume fraction (55%) of cuboidal coherent precipitates (average size ~90.8 nm, number density ~(7.2 ± 0.2) × 10$^{20}$ m$^{-3}$) from the expression:

$$S_{\text{specific-cuboidal}} = (6a^2)/(a^3) \qquad (11)$$

The resulting estimated value of $S_{\text{specific-cuboidal}}$ is $6.56 \times 10^7$ m$^{-1}$. Therefore, the high $S_{\text{specific}}$ for the plate-shaped precipitates and the associated elastic strain field in the A-MCA material are expected to maximize the interaction strength with dislocations, thereby achieving high strength.

## Estimation of the dislocation density

The dislocation density ($\rho$) in the current alloy system was quantified by using direct ECC imaging and indirect XRD methods. The value of $\rho$ for the D-MCA material with a homogeneous chemical distribution at the near-atomic scale was measured as $\rho = 2N_d/(L \cdot t)$, where $N_d$ is the intersection number with grids on the corresponding ECC images, $L$ is the total length of the grids, and $t$ is the probe depth. Multiple ECC images recorded under two-beam conditions were examined over a total area of $2.5 \times 10^6$ nm$^2$. The number density of dislocations in the D-MCA was measured to be $(3.0 \pm 0.3) \times 10^{15}$ m$^{-2}$. However, this method does not apply to MCAs that contain microstructural features with distinctive chemical differences. Therefore, $\rho$ was also calculated based on the Krivoglaz–Wilkens dislocation model[51], which provides a framework for analyzing dislocation-induced strain broadening and is used to determine dislocation density. XRD data were collected using Co K$\alpha$ radiation on a Bruker D8 advance diffractometer with a Lynxeye position-sensitive detector. The instrumental contribution was modeled using the fundamental parameter approach[51] via TOPAS software (NIST LaB6 standard). Crystalline size broadening was considered for the single fcc phase and fcc+L1$_2$ phases and was described as phenomenological Lorentzian-type broadening for simplicity (Supplementary Fig. 20). The dislocation densities for the D-MCA, A-MCA, and I-MCA materials were estimated to be $3.2 \times 10^{15}$, $2.5 \times 10^{15}$, and $2.8 \times 10^{15}$ m$^{-2}$, respectively. According to the Krivoglaz–Wilkens dislocation model, the calculated dislocation density values are comparable to the experimentally observed values.

## Estimation of yield strength and strengthening mechanism

Calculating different strengthening mechanisms and their coupling effects quantitatively and precisely is generally challenging. We estimated the yield strength ($\sigma_y$) of the A-MCA material by using the expression[52–54]:

$$\sigma_{y(P-MCA)} = \sigma_{base} + \sigma_{grain\ boundary} + \sigma_{dislocation + precipitation} \tag{12}$$

where $\sigma_{base} = 461.7$ MPa is the strengthening contribution of the lattice friction stress and the solid solution effect, $\sigma_{grain\ boundary}$ is the grain boundary strengthening, and $\sigma_{dislocation + precipitation}$ is the sum of the precipitation strengthening ($\sigma_{precipitation}$) and dislocation strengthening ($\Delta\sigma_{dislocation}$) according to the expression $\sigma_{dislocation + precipitation} = (\sigma_{dislocation}^2 + \sigma_{precipitation}^2)^{1/2}$ because the spacings of the dislocations and precipitates are very small[52,53]. Grain boundary strengthening was estimated from the classical Hall–Petch relationship as follows:

$$\Delta\sigma_{grain\ boundary} = k_y \cdot D^{-1/2} \tag{13}$$

where $k_y = 854.8$ MPa·μm$^{1/2}$ is the Hall–Petch slope adopted from the FeCoNiCr system[55] for a comparable grain size (~100 μm) to that of the current MCAs. The grain boundary contributions for the A-MCA and D-MCA materials were calculated to be 85.1 and 89.6 MPa, respectively.

A major contribution to the overall strength was dislocation strengthening because the A-MCA material experienced a high degree of cold deformation (60%) and short annealing at a relatively low temperature (10 min at 1073 K). More specifically, the dislocation number density increased significantly from $6 \times 10^{12}$ m$^{-2}$ under deformation-free condition[13] (B-MCA, as-homogenized) to $3.2 \times 10^{15}$ m$^{-2}$ under cold-rolled conditions (D-MCA). The dislocation density then decreased to $2.5 \times 10^{15}$ m$^{-2}$ for the A-MCA material (annealing at 1073 K for 10 min) containing metastable plate-like precipitates. The decrease in dislocation density in the A-MCA material was caused by the static recovery and rearrangement of dislocations during annealing, as this change was compensated by the formation of the matrix/precipitate interface. Dislocation strengthening is

calculated according to the classic Taylor model as follows:

$$\Delta\sigma_{dislocation} = M \cdot \alpha \cdot G \cdot b \cdot \rho^{\frac{1}{2}} \tag{14}$$

where $\alpha = 0.2$ is a constant for fcc metals. The dislocation strengthening of the A-MCA and D-MCA materials were evaluated to be 690.9 and 775.7 MPa, respectively.

Another major contribution is precipitation strengthening. For precipitates that are homogeneously distributed in the matrix, Orowan and cutting mechanisms are generally considered. Based on the experimental results (Fig. 3 and Supplementary Fig. 19), no Orowan looping was observed during deformation. In addition, the high number density, unique shape and pattern of the precipitates led to a small precipitate spacing that could not activate the Orowan mechanism[56]. Therefore, shearing is considered to be the dominant precipitation strengthening mechanism in the current A-MCA material. Considering that the precipitation cutting model is based on spherical precipitates, we first estimated that the precipitate shape was spherical. The radius ($r$) of the precipitates was calculated to be 15.8 nm based on the individual volume of the precipitates. The contributing factors to precipitate strengthening through cutting, including order strengthening ($\sigma_{ordering}$), modulus strengthening ($\sigma_{modulus}$) and coherency strengthening ($\sigma_{coherency}$), are expressed in the form[57]:

$$\sigma_{ordering} = 0.81M\left(\frac{\gamma_{APB}}{2b}\right)\left(\frac{3\pi f_v}{8}\right)^{1/2} \tag{15}$$

$$\sigma_{modulus} = 0.0055M(\Delta G)^{\frac{3}{2}}\left(\frac{2f_v}{G}\right)^{\frac{1}{2}}\left(\frac{r}{b}\right)^{\frac{3m}{2}-1} \tag{16}$$

$$\sigma_{coherency} = M \cdot \alpha_\varepsilon (G \cdot \varepsilon)^{3/2}\left(\frac{2r \cdot f_v}{G \cdot b}\right)^{1/2} \tag{17}$$

where $M = 3.06$ is the Taylor factor, b is the magnitude of the Burgers vector in the fcc matrix, $\alpha_\varepsilon \approx 2.6$ and $m = 0.85$ are constants, $\gamma_{APB} = 26$ mJ·m$^{-2}$ is the anti-phase boundary energy of the L1$_2$ precipitates achieved by DFT calculations, $\varepsilon \approx 2\delta/3$ is the constrained lattice misfit, the lattice misfit between the precipitates and matrix is given by the expression $\delta = 2(a_{L12} - a_{fcc})/(a_{L12} + a_{fcc})$, and $a_{L12}$ and $a_{fcc}$ are the lattice parameters of the L1$_2$ and fcc phases, respectively. A larger ($\sigma_{modulus} + \sigma_{coherency}$), which contributes prior to cutting, and $\sigma_{ordering}$, which contributes during cutting, are the determining factors for strengthening. Based on the above equations and calculations, the precipitation strengthening in the A-MCA material is ~2.74 GPa. This significant overestimation of precipitation strengthening results mainly from the large lattice misfit, which is due to the formation of partially formed hcp precipitates with relatively high distortions compared to those of the L1$_2$ precipitates. In addition, for precipitates with anisotropic morphologies other than spherical morphologies, it is challenging to estimate their precipitation contributions quantitatively because of the lack of appropriate estimations of the strengthening equations and interfacial strengthening[27]. Accordingly, we estimated the precipitation strengthening contribution in the A-MCA material based on the following expression:

$$\sigma_{y(P-MCA)} - \sigma_{y(B-MCA)} = \Delta\sigma_{grain\ boundary} + \sigma_{dislocation(P-MCA) + precipitation(P-MCA)}$$
$$- \sigma_{dislocation(B-MCA)} \tag{18}$$

where $\sigma_{y(B-MCA)} = \sigma_{base} + \sigma_{grain\ boundary(B-MCA)} + \sigma_{dislocation(B-MCA)}$. The sum of the different strengthening mechanisms in the D-MCA material is calculated to be ~1.33 GPa, which is very close to the experimentally measured value of ~1.32 GPa. $\sigma_{precipitation}$ and $\Delta\sigma_{dislocation + precipitation}$ for the A-MCA material are estimated to be 1.39 and 1.21 GPa, respectively.

The sum of the different strengthening mechanisms in the A-MCA material (1.94 GPa) is also very close to the experimental value (1.93 GPa). The above calculations clarify that the critical factors for the high yield strength of the A-MCA material are derived primarily from the combined effect of dislocations and precipitation strengthening.

The microstructure features and mechanical properties of the Fe-Co-Ni-Ta alloy variants with identical compositions but different thermomechanical processing methods are summarized in Supplementary Table 3.

## Data availability
The data that support the findings of this study are available on request from the corresponding author or the specify the conditions.

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

## Acknowledgements

L.H. acknowledges the China Scholarship Council (201906370028) and the Ernst Ruska-Centre (project number ER-C D-078). Z.L. acknowledges financial support by the Major Fundamental Research Program of Hunan Province (Grant No.: 2024JC0003). Synchrotron measurements were carried out at beamline P02.1, PETRA III of Deutsches Elektronen-Synchrotron (DESY, proposal number I-20211321). A.K. acknowledges the European Research Council under the European Union's Horizon 2020 Research and Innovation Programme (Grant No. 856538, project "3D MAGiC"). O.Gutfleisch is grateful for funding from the Deutsche Forschungsgemeinschaft (Project-ID 405553726, TRR270). The support of S.Nandy, A. Kwiatkowski da Silva, B. Breitbach, R.Zhou, and Y.Hu from the Max Planck Institute for Sustainable Materials is gratefully acknowledged.

## Author contributions

L.H. designed the research project; L.H., F.M., N.P., and A.K characterized the materials; L.H., J.W., and Y.W. analyzed the data; Y.Z., R. X., and H. Z. computed the DFT and ML results. R.S., Z.L., O.G., R.D., and D.R. conceptualized the paper. L.H. and D.R. wrote the paper. All authors contributed to the discussion.

## Funding

## Competing interests

The authors declare no competing interests.
