## [Transparent Peer Review file · Nature Communications]

Two-gigapascal-strong ductile soft magnets

Corresponding Author: Dr Liulu Han

Version 0:

Reviewer comments:

Reviewer #1

(Remarks to the Author)

This is an in-depth study reporting concurrent achievements of strength, ductility, and magnetic performance in Fe-Ni-Co-Ta based multicomponent alloys (MCAs) as well as the underlying microstructure-based mechanisms responsible for these achievements. While the excellent combined mechanical properties (i.e., strength and ductility) are impressive, the critical needs for such properties are not well motivated. For example, one would imagine such a high strength could make fabrication/shape-forming of these SMMs more difficult. Namely, for SMMs to be useful, maybe a lower strength (i.e., 1 GPa or lower) is better than a ~2 GPa strength.

Several minor comments below:

1. There are no page numbers on either of the documents.
2. The authors performed extended isothermal treatment (1073K, 60 to 300 min), and reported that the formation of incoherent phase boundaries and coarsening of precipitates results in an undesired decrease in strength and an increase in coercivity. Since thermal stability of the coherent precipitates is a concern for the long-term performance of these SMMs, it would be very useful if long-term (maybe 1000h or longer) aging investigation is performed at expected operating temperatures of these SMMs.
3. Selected area electron diffraction (SEAD) was used to infer that the coherent L12 nanoprecipitates are the "major fraction". However, SAED only examines a very small volume of the sample material. XRD or beamline-based method covering a larger volume would be more convincing.

Reviewer #2

(Remarks to the Author)

Dear Co-authors,
Dear Editors,

the authors report on the design and characterization of a bulk multicomponent soft magnetic material with remarkable mechanical and magnetic properties, including strength of two-gigapascal, good ductility (12%) and saturation magnetization (115 Am²·kg⁻¹), low coercivity (0.36 kA·m⁻¹) and high electrical resistivity (62.5404 μΩ·cm). Furthermore, the authors systematically explored strengthening and toughening mechanisms in this material using advanced characterization methods such as high resolution TEM and atom probe analysis. The manuscript itself is very well structured and very well written. The data are of the highest quality and are clearly presented. The manuscript can be published in its current state.

Although the manuscript is of high importance and interest, it is part of a series of publications by the same co-authors. Three manuscripts from this series are listed below:

- "A mechanically strong and ductile soft magnet with extremely low coercivity" published in Nature
- "Strong and ductile high temperature soft magnets through Widmanstätten precipitates" published in Nature Communications
- "Ultrastrong and Ductile Soft Magnetic High-Entropy Alloys via Coherent Ordered Nanoprecipitates" in Advanced Materials

In the previous publications, the authors report on the same material (in terms of chemical composition) and similar magnetic properties. The main difference is an improved strength which was achieved by an extra cold rolling step.

In my personal opinion, the novelty of the current manuscript does not meet the level of novelty expected for publications in Nature Communications. Therefore, I cannot recommend this excellent manuscript for publication in Nature Communications.

Reviewer #3

(Remarks to the Author)

The authors designed a bulk HEA SMM with unprecedented mechanical strength and good ductility (12%) by engineering anisotropic and coherent nanoprecipitates with sizes less than the magnetic domain wall width. The author provided details on the magnetic properties' mechanism from experimental and computational perspectives. The key result of this work is that the mechanical strengthening mechanisms do not interfere with the magnetic domain wall motion, thus bypassing the traditional constraint (dilemma) that whatever the mechanism making the SMM mechanically hard will make it magnetically hard. However, the experimentally measured coercivity, 360 A/m, is not low. Actually, it is one order of magnitude higher than that of the Hiperco 50, which is about 40 A/m after standard magnetic annealing. Assuming nanoprecipitates have no effects on magnetic domain pinning, then what is the mechanism for FeCoNiTa exhibiting such high coercivity? If author can explain the high coercivity is from the matrix, then the conclusion of small nanoprecipitates will not affect magnetic properties is valid.

In addition, in line 268, the author claims the new alloy's resistivity is better than Fe-Si. This statement ignored 6.5%Si steel, whose resistivity is about 80 $\mu\text{Ohm-cm}$. Lastly, the author used the word "engine" several times in the manuscript. An engine converts fuel to mechanical energy, while a motor converts electricity to mechanical energy. SMM is more for motor applications, not for engine applications.

Version 1:

Reviewer comments:

Reviewer #2

(Remarks to the Author)

Dear Editor,
Dear Authors,

I am not convinced with the comments to my original review.

I still believe that this excellent publication is one out of currently four publications by the same authors reporting on the same material. Additional cold rolling treatment induced to the well-studied alloy reported in the previous three publications (by the same co-authors) resulted in the expected strengthening effect. I cannot recommend this excellent work for publication in Nature Communication.

Sincerely,
Reviewer

Reviewer #3

(Remarks to the Author)

The work by itself does have its value, despite that fact that coercivity is higher than most of the competing SMM. It is better to state this fact than claiming "good" magnetic properties. Afterall, some applications may care more mechanical strength and are willing to take a hit at coercivity. I do suggest accepting this paper for publication under the condition that a proper acknowledgement of the relatively high coercivity.

Version 2:

Reviewer comments:

Reviewer #3

(Remarks to the Author)

The revision has addressed my concerns on the tradeoff between mechanical and magnetic properties.

RESPONSE TO REVIEWERS' COMMENTS

We cordially thank the reviewers for their valuable suggestions and comments on our manuscript. Our response is structured as follows: The comments from the reviewers are copied below in black and italic font. For each comment, we present a response item and the corresponding manuscript modifications (blue font). The changes in the amended manuscript are highlighted in yellow.

Specific response to reviewer concern (1: is there a need for such materials?)

The reviewers agree that the main goal of this study – to increase yield strength and ductility without sacrificing magnetic properties – has been well achieved. Yet, the question remains if such mechanical properties are indeed important in soft magnets.

The clear answer, also covered by the works cited below, is ‘yes’:

Any electrical equipment, transformer, or motor designed necessarily with SMMs is generally subject to a complex mechanical load exposure profile. The challenge of electrification is that the corresponding components are subjected to high manufacturing-related, dynamic, and alternating mechanical loads¹. These parts are currently the most vulnerable components in electric automobiles but are mechanically very soft. SMMs with high yield strength are targeted since the yield strength is where irreversible plastic deformation starts. Therefore, loading stresses must remain below this value for the safe and efficient operation of magnetic parts.

Another severe extrinsic cause of the loss of soft magnetic performance under mechanical loading is the manufacturing and processing of the corresponding parts, i.e., the many complex manufacturing steps to which the materials are exposed. These trimming and forming processes are associated with high mechanical loads, either exposed homogeneously across the bulk sheet or locally. These severe manufacturing-related loads, when applied to mechanically soft (or brittle) magnetic materials, create a cosmos of lattice defects (dislocations, cracks, delamination, etc.) in the material, which pin the otherwise highly mobile magnetic domains and thus contribute to the loss of magnetic properties. For example, Fe-Co alloys are not widely used due to their brittleness and inherent low electrical resistivity².

A multifunctional SMM containing high yield stress (reversible deformation at high mechanical load), good ductility (enabling manufacturing and reserve as a safety margin against fracture), good soft magnetic properties, and high electrical resistivity are needed.

Specific response to reviewer concern (2: does the here suggested approach constitute a novel design concept?):

Yes, it does.

Our new multicomponent magnet design approach allows us to make and tune materials covering a much broader spectrum of property combinations, including excellent mechanical strength, good ductility and damage tolerance, good soft magnetic properties, and electrical resistivity.

The challenge is – as mentioned in the article – that the soft magnetic components of these electrical components are mechanically much softer than the other components used in the construction of machines and automobiles and are, therefore, already plastically stressed at much lower loads, both in operation and during manufacturing and processing, and thus lose their magnetic properties, which leads directly and linearly to losses in the efficiency of the electrical drives and the transformers.

However, attempts have been made to improve yield strength without sacrificing ductility and soft magnetic properties do not work. For example, coherent cuboidal precipitates in well-tuned size can result in low magnetic domain wall pinning. However, the increase in attainable strength has proven to be limited to values far below one gigapascal³. The lack of ductility, which leads to the catastrophic failure of soft magnetic bulk metallic glasses, is the most critical long-standing bottleneck that prohibits their wider commercial applications for such soft magnetic parts^{4,5}. Enhancing the room-temperature ductility of bulk metallic glasses has been a massive research trend in recent decades. It was found that practically all measures that enhance ductility inevitably deteriorate their soft magnetic properties and strength.

We present a generic solution based on a multicomponent nanostructuring design strategy using morphologically-anisotropic and coherent ferromagnetic nanoprecipitates. The synergistic effect of the morphology, distribution, size, and composition of the precipitates is guided by the following rules: (a) The anisotropic morphology increases the total surface area per unit of volume and lattice mismatch, thereby strengthening the material through mechanical interactions with dislocations. (b) Through a finer size distribution, the effective interprecipitate spacing is decreased, enhancing strengthening. (c) The micromagnetically relevant length scales of the precipitates remain below the magnetic domain wall width to leave domain wall motion unaffected, thus reducing magnetic losses. (d) High concentrations of strong ferromagnetic elements with the highest possible magnetization and moderate costs are used. (e) We solved the design challenge of triggering coherent, ferromagnetic, and anisotropic nanoprecipitates in thermodynamically nonequilibrium conditions while suppressing capillary-driven coarsening and transitioning to full incoherency.

The precipitation is triggered by dislocations from preceding deformation during heat treatment via the introduced plastic microbands (via plastic deformation) of planar dislocations along the {111} crystallographic planes to control the density and morphology. All dimensions of the precipitates need to be controlled below the magnetic domain wall width to avoid significant interactions between the precipitates and magnetic domain wall width for low coercivity. This leads to thermodynamically nonequilibrium conditions for precipitates with an anisotropic morphology and fine distribution; thus, the precipitates have a large specific surface area, small interprecipitate spacing, and high lattice mismatch. With this new design strategy, we double the current yield strength of established soft magnetic at maintained ductility.

Reviewers' comments:

Reviewer #1

This is an in-depth study reporting concurrent achievements of strength, ductility, and magnetic performance in Fe-Ni-Co-Ta based multicomponent alloys (MCAs) as well as the underlying microstructure-based mechanisms responsible for these achievements. While the excellent combined mechanical properties (i.e., strength and ductility) are impressive, the critical needs for such properties are not well motivated. For example, one would imagine such a high strength could make fabrication/shape-forming of these SMMs more difficult. Namely, for SMMs to be useful, maybe a lower strength (i.e., 1GPa or lower) is better than a ~2GPa strength.

Reply:

Thank you for recognizing the validity of our conclusions and bringing up the pertinent point. Indeed, we agree that the critical needs for such properties are important. As we mentioned in the specific response to the reviewer's concern (1: is there a need for such materials for the requirement of such a property profile?), such multifunctional SMM containing high yield stress (reversible deformation at high mechanical load), good ductility (enabling manufacturing and reserve as a safety margin against fracture), good soft magnetic properties and high electrical resistivity is needed.

For the fabrication and shape-forming of these SMMs, the initial homogenized sample (B-MCA) without cold rolling shows a low yield strength (~ 0.5 GPa), high ductility ($>50\%$), and, therefore, good manufacturing and processing performance. After simple thermomechanical treatment, namely, cold rolling for a 60% thickness reduction and annealing at 800°C for 10 min, the yield strength improved almost four times (1.9 GPa) while remaining ductile ($>10\%$). Therefore, this material can potentially be applied to complex processing.

Several minor comments below:

1. There are no page numbers on either of the documents.

Reply and modifications:

Thank you for highlighting this mistake. In the revised version, we have added page numbers to all the documents.

2. The authors performed extended isothermal treatment (1073K, 60 to 300 min), and reported that the formation of incoherent phase boundaries and coarsening of precipitates results in an undesired decrease in strength and an increase in coercivity. Since thermal stability of the coherent precipitates is a concern for the long-term performance of these SMMs, it would be very useful if long-term (maybe 1000h or longer) aging investigation is performed at expected operating temperatures of these SMMs.

Reply:

Thank you for bringing up this very important point. We fully agree that the thermal stability of coherent precipitates is a concern for long-term performance and applications, especially considering that we contain a high density of thermodynamically nonequilibrium nanoprecipitates. Considering that our target application temperature is at room temperature and the corresponding diffusion/growth kinetics of the precipitates are extremely slow and negligible, we expect the samples to be stable.

3. Selected area electron diffraction (SEAD) was used to infer that the coherent $L1_2$ nanoprecipitates are the “major fraction”. However, SAED only examines a very small volume of the sample material. XRD or beamline-based method covering a larger volume would be more convincing.

Reply:

We fully agree with your suggestions and comply.

Compared to high-resolution TEM characterization and fast Fourier transform (FFT) analysis (tens of nanometers in range), SAED analysis (Extended data Fig. 5) covers more area (1-5 μm). It provides more statistical analysis containing a larger area. However, the SAED results indicate a relatively small volume compared to the bulk volume.

After checking our XRD (including refined) results, as shown in Extended Data Fig. 2 and Extended Data Fig. 20, we can only identify $L1_2$ nanoprecipitates. This finding also supports that the volume fraction of the $L1_2$ nanoprecipitates is the major fraction, and the fraction of the ordered hcp is less than 5%.

Modifications:

Please see Extended Data Fig. 20: “No peaks of ordered hcp are detected, indicating that the $L1_2$ nanoprecipitates constitute the major fraction of the A-MCA.”

Reviewer #2 (Remarks to the Author):

Dear Co-authors,

the authors report on the design and characterization of a bulk multicomponent soft magnetic material with remarkable mechanical and magnetic properties, including strength of two-gigapascal, good ductility (12%) and saturation magnetization (115 $\text{Am}^2\text{-kg}^{-1}$), low coercivity (0.36 $\text{kA}\cdot\text{m}^{-1}$) and high electrical resistivity (62.5404 $\mu\Omega\cdot\text{cm}$). Furthermore, the authors systematically explored strengthening and toughening mechanisms in this material using advanced characterization methods such as high

resolution TEM and atom probe analysis. The manuscript itself is very well structured and very well written. The data are of the highest quality and are clearly presented. The manuscript can be published in its current state.

Reply: Thank you for appreciating the scientific quality of our work. We are delighted to have the opportunity to strengthen our case by revising the manuscript.

Although the manuscript is of high importance and interest, it is part of a series of publications by the same co-authors. Three manuscripts from this series are listed below:

- *"A mechanically strong and ductile soft magnet with extremely low coercivity" published in Nature*
 - *"Strong and ductile high temperature soft magnets through Widmanstätten precipitates" published in Nature Communications*
 - *"Ultrastrong and Ductile Soft Magnetic High-Entropy Alloys via Coherent Ordered Nanoprecipitates" in Advanced Materials*
- In the previous publications, the authors report on the same material (in terms of chemical composition) and similar magnetic properties.*

Reply:

We thank you for this general comment and highly appreciate the input, but we share different opinions. We also list our main design strategy and findings below. We have emphasized the design strategy in the manuscript to highlight this novelty.

In our previous publications, only the "Strong and ductile high-temperature soft magnets through Widmanstätten precipitates" share the same chemical composition as the current one. However, our previous work focused on high-temperature materials with different research directions, design strategies, microstructures, and mechanisms, i.e., combinations of thermal stability and soft magnetism, for potential high-temperature applications.

The other published references you mentioned are all based on thermodynamic equilibrium precipitates, such as coherent nanoprecipitates with cuboidal shapes. This is generally reported for different material systems. However, motivated by the increasing demand for new SMMs that are able to withstand harsh and dynamic mechanical loads in electrical engines and high-speed flywheel energy storage systems, multifunctional profiles, i.e., those with high yield strength and good ductility, are needed to avoid irreversible plastic deformation during application without sacrificing soft magnetic performance. However, the yield strengths of soft magnetic materials are currently far below one gigapascal. Most methods that enhance strength often result in embrittlement and introduce stress fields that can pin magnetic domains. This motivated our work and led to the new design strategy and mechanism listed below.

The main difference is an improved strength which was achieved by an extra cold rolling step.

Reply: For the first time, we introduced morphologically anisotropic, thermodynamically nonequilibrium, and coherent ferromagnetic nanoprecipitates in soft magnetic materials as a new design strategy. The anisotropic morphology increases the specific surface area (total surface area per unit of volume) and lattice mismatch, thereby strengthening the material through mechanical interactions with dislocations. We introduced plastic microbands (via plastic deformation) of planar dislocations along the {111} crystallographic planes to control the density and morphology to achieve such nanostructures under thermodynamically nonequilibrium conditions. The micromagnetically relevant length scales of the precipitates remain below the magnetic domain wall width to ensure that domain wall motion is unaffected, thus maintaining the soft magnetic properties. With this new design strategy, we double the current yield strength of established soft magnetic materials while maintaining ductility.

In my personal opinion, the novelty of the current manuscript does not meet the level of novelty expected

for publications in Nature Communications. Therefore, I cannot recommend this excellent manuscript for publication in Nature Communications.

Reply:

In addition to the points mentioned in the specific response to reviewer concern (1: is there a need for such materials?) and Specific response to reviewer concern (2: does the here suggested approach constitute a novel design concept?), we also designed new methodologies in this work. This includes computational and experimental methods. More specifically, correlative microstructure and magnetic feature analysis from the microscale to the nanoscale has been performed. We developed a computational simulation method based on atomic-scale experiments and studied it via machine learning. These methods were all developed for the first time and have been applied in both the high-entropy material and magnetic communities.

Modifications:

We wrote several sentences to highlight the novelty:

See: “A higher yield strength of SMMs is thus essential for preventing the loss of magnetic performance and component failure due to plastic deformation. However, the yield strengths of SMMs are today far below one gigapascal.”,

and “...Our approach introduces morphologically anisotropic and coherent nanoprecipitates in an iron–nickel–cobalt–tantalum multicomponent alloy ...”,

and “...Precipitation is triggered by dislocations from preceding deformation during heat treatment, and all dimensions of the precipitates need to be controlled below the magnetic domain wall width to avoid significant interactions between the precipitates and magnetic domain wall width for low coercivity. This leads to thermodynamically nonequilibrium conditions for precipitates with an anisotropic morphology (long edge of ~53 nm, width of ~8 nm) and fine distribution (number density of $\sim 2.3 \times 10^{22} \cdot \text{m}^{-3}$); thus, the precipitates have a large specific surface area ($3.3 \times 10^8 \text{ m}^{-1}$), small interprecipitate spacing (26 nm), and high lattice mismatch (1.3%)...”.

and “Here, we propose a novel strategy to resolve this dilemma by means of a nanostructuring approach using morphologically-anisotropic ferromagnetic nanoprecipitates. The synergistic effect of the morphology, distribution, size and composition of the precipitates is guided by the following rules: (a) The anisotropic morphology increases the total surface area per unit of volume and lattice mismatch, thereby strengthening the material through mechanical interactions with dislocations. (b) Through a finer size distribution, the effective interprecipitate spacing is decreased, enhancing strengthening. (c) The micromagnetically relevant length scales of the precipitates remain below the magnetic domain wall width to leave domain wall motion unaffected, thus reducing magnetic losses¹⁵. (d) High concentrations of strong ferromagnetic elements with the highest possible magnetization and moderate costs are used. (e) We solved the design challenge of triggering coherent, ferromagnetic, and anisotropic nanoprecipitates under thermodynamically nonequilibrium conditions while suppressing capillary-driven coarsening and transitioning to full incoherency.”

Reviewer #3:

The authors designed a bulk HEA SMM with unprecedented mechanical strength and good ductility (12%) by engineering anisotropic and coherent nanoprecipitates with sizes less than the magnetic domain wall width. The author provided details on the magnetic properties' mechanism from experimental and computational perspectives. The key result of this work is that the mechanical strengthening mechanisms do not interfere with the magnetic domain wall motion, thus bypassing the traditional constraint (dilemma) that whatever the mechanism making the SMM mechanically hard will make it magnetically hard. However, the experimentally measured coercivity, 360 A/m, is not low. Actually, it is one order of magnitude higher than that of the Hiperco 50, which is about 40 A/m after standard magnetic annealing. Assuming nanoprecipitates have no effects on magnetic domain pinning, then what is the mechanism for FeCoNiTa exhibiting such high coercivity? If author can explain the high coercivity is from the matrix, then the conclusion of small nanoprecipitates will not affect magnetic properties is valid.

Reply:

Thank you for this great observation. However, according to magneto-optical Kerr effect (MOKE) microscopy, nanoprecipitates do not pin the movement of magnetic domain walls as grain boundaries do. We believe that nanoprecipitation introduces additional internal elastic stress fields due to its high number density and different lattice distortions compared to those of the matrix. This can lead to relatively high coercivity. However, it is difficult to differentiate the effects of the matrix and nanoprecipitates using this method due to limitations in optical microscopy. Further characterization and analysis down to the nanoscale from the micromagnetic simulation (Extended Data Fig. 17) indicated that introducing nanoprecipitates results in an exchange interaction with the matrix, indicating a pinning effect at the nanoscale. In addition, the intrinsic properties of the matrix are improved after precipitation, i.e., the matrix contains 95% and 97% ferromagnetic elements before and after precipitation, respectively.

In addition, in line 268, the author claims the new alloy's resistivity is better than Fe-Si. This statement ignored 6.5%Si steel, whose resistivity is about 80 uOhm-cm.

Reply:

Thank you for this great suggestion. We modified this statement in the revised manuscript.

Modifications:

Please see: “This comparison shows that the ρ_c value of the new A-MCA material ($62.5 \mu\Omega\cdot\text{cm}$) is higher than the corresponding reference values for all the pure metals, Fe-Si (Si content < 6.5 wt.%),²¹, Fe-Co²², comparable to Fe-Ni³¹ alloys and established MCAs^{25,28,32,33}.”

Lastly, the author used the word “engine” several times in the manuscript. An engine converts fuel to mechanical energy, while a motor converts electricity to mechanical energy. SMM is more for motor applications, not for engine applications.

Reply:

Thank you for your careful reading and for bringing this point up. We modified this statement in the revised manuscript.

Modifications:

Please see the abstract: “However, SMMs in electric motors experience increasingly harsh mechanical loads due to high rotational speeds.”, and introduction: “...are able to withstand harsh and dynamic mechanical loads in electrical motors and high-speed flywheel energy storage system...”, and conclusion: “increasing demand for SMM components in electrical vehicle motors and high-speed flywheel energy storage devices...”

References

- [1] R. S. Sundar, S. C. Deevi Int. Mater. Rev. 50, 157–192 (2005). <https://doi.org/10.1179/174328005X14339>
- [2] C. W. Chen, Magnetism and metallurgy of soft magnetic materials (Courier Corporation, 2013). ISBN-0-486-64997-0
- [3] Han, L., Maccari, F., Souza Filho, I.R. et al. A mechanically strong and ductile soft magnet with extremely low coercivity. Nature 608, 310–316 (2022). <https://doi.org/10.1038/s41586-022-04935-3>
- [4] T. F. Babuska et al., Acta Mater. 180, 149–157 (2019). <https://doi.org/10.1016/j.actamat.2019.08.044>
- [5] M. Garibaldi et al., Acta Mater. 110, 207–216 (2016). <https://doi.org/10.1016/j.actamat.2016.03.037>

RESPONSE TO REVIEWERS' COMMENTS

Manuscript Title: Two-gigapascal-strong ductile soft magnets

Manuscript number: NCOMMS-24-30250A-Z

We thank the reviewers for the valuable suggestions and comments on our manuscript. Our response is structured as follows: The comments from the reviewers are copied below in black and italic font. For each comment, we present a response item and the corresponding manuscript modifications (**green font**). The changes in the amended manuscript are highlighted in **yellow**.

Reviewer #2:

Dear Authors,

I am not convinced with the comments to my original review.

I still believe that this excellent publication is one out of currently four publications by the same authors reporting on the same material. Additional cold rolling treatment induced to the well-studied alloy reported in the previous three publications (by the same co-authors) resulted in the expected strengthening effect. I cannot recommend this excellent work for publication in Nature Communication.

Sincerely,

Reviewer

Reply:

Thank you for your thoughtful comments and for acknowledging the quality of our work. We deeply appreciate your feedback. We respect your concerns and will carefully consider your feedback in our future work.

Reviewer #3 (Remarks to the Author):

The work by itself does have its value, despite that fact that coercivity is higher than most of the competing SMM. It is better to state this fact than claiming "good" magnetic properties. After all, some applications may care more mechanical strength and are willing to take a hit at coercivity. I do suggest accepting this paper for publication under the condition that a proper acknowledgement of the relatively high coercivity.

Reply:

Thank you very much for your valuable feedback and for recommending our paper for publication. We appreciate your acknowledgment of the work's value and your constructive suggestions.

We fully agree with you that our materials have a slightly higher coercivity compared to other conventional SMMs. As you correctly mentioned, certain applications can prioritize good mechanical properties over extremely low coercivity, manifesting a certain trade-off situation, depending on the targeted use case. In compliance with this comment we revised our manuscript to explicitly clarify our relatively high coercivity, as our main focus is to propose

new material design strategy to achieve a good combination of ultrahigh mechanical strength and ductility in SMMs.

Response:

We revised the manuscript accordingly to ensure that our claims about "good" magnetic properties are more precise, and we have also included further discussion on how the coercivity of the current materials compare to other SMMs. We hope that this revision will directly address your comments by acknowledging the higher coercivity and framing it as an additional consideration for certain applications.

Please see revised manuscript.

1. Please see modification of the abstract:

“...The material has a yield strength of 1.9 GPa, elongation of 12.6%, and multi-functional performance. This nanostructuring approach offers a pathway to ultrastrong and ductile SMMs with moderately increased coercivity that can be tolerated in exchange for significantly improved mechanical performance for sustainable electrification.”

2. please see Results and discussion:

Page 11:

“...The A-MCA material has an extremely high value of σ_y with moderate H_c ...”

“...It should be noted that although the coercivity of the current A-MCA is higher than that of some conventional SMMs, this is acceptable for applications under high mechanical stresses that prioritize high mechanical strength over minimal coercivity...”

Page 16:

“...It is worth noting that the coercivity values of the current materials are slightly higher compared to those of conventional SMMs. Nevertheless, the excellent mechanical properties, i.e., ultrahigh yield strength paired and good ductility make it suitable for applications under severe mechanical stresses where mechanical robustness outweighs the need for minimal coercivity..”

3. please see conclusion:

Page 19:

“...Although introducing a high number density and different types of lattice defects results in a moderately higher coercivity, this new material design concept provides a novel approach for highly efficient electromechanical energy conversion applications where mechanical robustness is prioritized over minimal coercivity. It is particularly suited for highly mechanically stressed magnetic parts exposed to severe centrifugal mechanical loads during service or manufacturing, for which conventional soft magnetic materials are mechanically too brittle or too soft....”